# Historical (1960–2014) lightning and LNO$_x$ trends and their controlling factors in a chemistry–climate model

Yanfeng He[1], Kengo Sudo[1,2]

[1] Graduate School of Environment Studies, Nagoya University, Nagoya, 464-8601, Japan

[2] Japan Agency for Marine–Earth Science and Technology (JAMSTEC), Yokohama, 237-0061, Japan

*Correspondence to*: Yanfeng He (hyf412694462@gmail.com)

**Abstract.** Lightning can cause natural hazards that result in human and animal injuries or fatalities, infrastructure destruction, and wildfire ignition. Lightning-produced NO$_x$ (LNO$_x$), a major NO$_x$ (NO$_x$=NO+NO$_2$) source, plays a vital role in atmospheric chemistry and global climate. The Earth has experienced marked global warming and changes in aerosol and aerosol precursor emissions (AeroPEs) since the 1960s. Investigating long-term historical (1960–2014) lightning and LNO$_x$ trends can provide important indicators for all lightning-related phenomena and for LNO$_x$ effects on atmospheric chemistry and global climate. Understanding how global warming and changes in AeroPEs influence historical lightning–LNO$_x$ trends can be helpful in providing a scientific basis for assessing future lightning–LNO$_x$ trends. Moreover, global lightning activities' responses to large volcanic eruptions such as the 1991 Pinatubo eruption are not well elucidated, and are worth exploring. This study employed the widely used cloud top height lightning scheme (CTH scheme) and the newly developed ice-based ECMWF-McCAUL lightning scheme to investigate historical (1960–2014) lightning–LNO$_x$ trends and variations and their influencing factors (global warming, increases in AeroPEs, and Pinatubo eruption) in the framework of the CHASER (MIROC) chemistry–climate model. Results of sensitivity experiments indicate that both lightning schemes simulated almost flat global mean lightning flash rate anomaly trends during 1960–2014 in CHASER (Mann-Kendall trend test (significance inferred as 5%) shows no trend for the ECMWF-McCAUL scheme, but a 0.03 % yr$^{-1}$ significant increasing trend is detected for the CTH scheme). Moreover, both lightning schemes suggest that past global warming enhances historical trends of global mean lightning density and global LNO$_x$ emissions in a positive direction (around 0.03% yr$^{-1}$ or 3% K$^{-1}$). However, past increases in AeroPEs exert an opposite effect to the lightning–LNO$_x$ trends (-0.07% yr$^{-1}$ – -0.04% yr$^{-1}$ for lightning and -0.08% yr$^{-1}$ – -0.03% yr$^{-1}$ for LNO$_x$) when one considers only the aerosol radiative effects in the cumulus convection scheme. Additionally, effects of past global warming and increases in AeroPEs on lightning trends were found to be heterogeneous across different regions when analyzing lightning trends on the global map. Lastly, this report is the first of study results suggesting that global lightning activities were suppressed markedly during the first year after the Pinatubo eruption shown in both lightning schemes (global lightning activities decreased by as much as 18.10% simulated by the ECMWF-McCAUL scheme). Based on simulated suppressed lightning activities after the Pinatubo eruption, findings also indicate that global LNO$_x$ emissions decreased after the 2–3-year Pinatubo eruption (1.99%–8.47% for the annual percentage reduction). Model intercomparisons of lightning flash rate trends and variations between our study (CHASER) and other

Coupled Model Intercomparison Project Phase 6 (CMIP6) models indicate great uncertainties in historical (1960–2014)
global lightning trend simulations. Such uncertainties must be investigated further.
**1 Introduction**
Lightning, an extremely energetic natural phenomenon, occurs at every moment somewhere on Earth: its average occurrence
frequency is approximately 46 times per second (Cecil et al., 2014). Lightning generation is associated with electric charge
separation, which is mainly realized by collisions between graupel and hail and hydrometeors of other types within
convective clouds (Lopez, 2016). As a natural hazard, lightning can cause human and animal injuries and fatalities,
infrastructure destruction, and wildfire ignition (Cerveny et al., 2017; Cooper and Holle, 2019; Jensen et al., 2022;
Veraverbeke et al., 2022). Lightning-produced $NO_x$ ($LNO_x$) accounts for around 10% of the global tropospheric $NO_x$
($NO_x=NO+NO_2$) source. It is regarded as the dominant $NO_x$ source in the middle to upper troposphere (Schumann and
Huntrieser, 2007; Finney et al., 2016b). Moreover, $LNO_x$ plays a crucially important role in atmospheric chemistry and
global climate by affecting the abundances of OH radical, important greenhouse gases (GHGs) such as ozone and methane,
and other trace gases (Labrador et al., 2005; Schumann and Huntrieser, 2007; Wild, 2007; Liaskos et al., 2015; Finney et al.,
2016a; Murray, 2016; Tost, 2017; He et al., 2022b).

Reportedly, the lightning flash rate (LFR) is related to the stage of convective cloud development (Williams et al., 1989),
Convective Available Potential Energy (CAPE) (Romps et al., 2014), cloud liquid–ice water content (Saunders et al., 1991;
Finney et al., 2014) and even to the convective precipitation volume (Goodman et al., 1990; McCaul et al., 2009; Romps et
al., 2014). Long-term global warming is associated with changes in the overall temperature and relative humidity profiles in
the atmosphere and global convective adjustment (Manabe and Wetherald, 1975; Del Genio et al., 2007), which can strongly
affect the lightning-related factors described above. Consequently, long-term global warming can be a fundamentally
important factor affecting long-term variations in global lightning activity. Findings from many earlier numerical simulation
studies manifest that global lightning activities are sensitive to long-term global warming, with most studies showing 5–16%
(average around 10%) increases in global lightning activities per 1 K global warming (Price and Rind, 1994; Zeng et al.,
2008; Hui and Hong, 2013; Banerjee et al., 2014; Krause et al., 2014; Romps et al., 2014; Clark et al., 2017). However, other
numerical simulation studies such as those using an ice-based lightning scheme or convective mass flux as a proxy to
parameterize lightning have yielded opposite results, suggesting that global lightning activity will decrease under long-term
global warming (Clark et al., 2017; Finney et al., 2018).

Aside from long-term global warming, changes in aerosol loading can also be responsible for long-term global lightning
activity variations. Aerosols influence lightning activity through aerosol radiative and microphysical effects, but the degree
to which the two distinct effects influence regional or global scale lightning activities remains unclear (Yuan et al., 2011;
Yang et al., 2013; Tan et al., 2016; Altaratz et al., 2017; Wang et al., 2018; Liu et al., 2020). Further research is needed. It is
urgently necessary to elucidate the effects of aerosol radiative and microphysical effects on lightning on a global scale. The
aerosol radiative effects indicate that aerosols can heat the atmospheric layer and can cool the Earth's surface by absorbing
and scattering solar radiation (Kaufman et al., 2002; Koren et al., 2004, 2008; Li et al., 2017). Thereby, convection and
electrical activities are likely to be inhibited (Koren et al., 2004; Yang et al., 2013; Tan et al., 2016). The microphysical
effects suggest that by acting as cloud condensation nuclei (CCN) or as ice nuclei, aerosols can reduce the mean size of
cloud droplets, consequently suppressing the coalescence of cloud droplets into raindrops. As a result, more liquid water
particles are uplifted to higher mixed-phase regions of the troposphere, where they invigorate lightning (Wang et al., 2018;
Liu et al., 2020).

The Earth has experienced a considerable degree of global warming and changes in AeroPEs since the 1960s (Hoesly et al.,
2018; Climate at a Glance | National Centers for Environmental Information (NCEI), 2022). However, how historical
lightning has trended and how lightning has responded to historical global warming and changes in AeroPEs are not well
examined. This topic is worth exploring because historical lightning densities are indicators for all lightning-related
phenomena (Price and Rind, 1994). Exploring the historical global $LNO_x$ emission trend is also meaningful because it can
indicate the effects of $LNO_x$ emissions on atmospheric chemistry and global climate. Furthermore, investigating the effects
of historical global warming and increases in AeroPEs on historical lightning–$LNO_x$ trends can provide a basis for assessing
future lightning–$LNO_x$ trends.

Large-scale volcanic eruptions such as the 1991 Pinatubo eruption inject tremendous amounts of sulfuric gas into the
stratosphere, where it converts to $H_2SO_4$ aerosols. Consequently, the stratospheric aerosols have increased in abundance after
the volcanic eruptions. The enhanced stratospheric aerosol layer can cool the Earth's surface heterogeneously and can
decrease the total amount of water in the atmosphere (Soden et al., 2002; Boucher, 2015, p.63). The near-global
perturbations in the radiative energy balance and meteorological fields caused by such strong volcanic eruptions might
influence global lightning activities. If so, there might be ramifications for all lightning-related phenomena. Nevertheless,
they remain poorly understood.

In our earlier work, we developed a new process and ice-based lightning scheme called the ECMWF-McCAUL scheme (He
et al., 2022b). This lightning scheme was developed by combining benefits of the lightning scheme used in the European
Centre for Medium-Range Weather Forecasts (ECMWF) forecasting system (Lopez, 2016) and those presented in reports by
McCaul et al. (McCaul et al., 2009). The ECMWF-McCAUL scheme simulated the best lightning density spatial
distributions among four existing lightning schemes when compared against satellite lightning observations (Lightning
Imaging Sensor (LIS) and Optical Transient Detector (OTD)) during 2007–2011. The sensitivity of global lightning activity
to changes in surface temperature on a decadal timescale was estimated as 10.13% K$^{-1}$ using the ECMWF-McCAUL scheme
(He et al., 2022b), which is close to most past estimates (average around 10% K$^{-1}$).

Using a chemistry–climate model CHASER (MIROC) with two lightning schemes (the widely used cloud top height scheme
and the ice-based ECMWF-McCAUL scheme), we investigated historical lightning–LNO$_x$ trends quantitatively and
ascertained how global warming, increases in AeroPEs, and the Pinatubo eruption respectively influenced them. Using two
lightning schemes, we demonstrated the sensitivities of different lightning schemes to historical global warming, increases in
AeroPEs, and the Pinatubo eruption.

Research methods including the model description and experiment setup, are described in Sect. 2. In Sect. 3.1, the simulated
historical lightning distributions and trends are validated using LIS/OTD lightning observations. Section 3.2 presents the
effects of global warming and increases in AeroPEs on historical lightning–LNO$_x$ trends. In Sect. 3.3, the Pinatubo volcanic
eruption effects on historical lightning–LNO$_x$ trends are discussed. Section 3.4 elucidated model intercomparisons of LFR
trends and variation between our study (CHASER) and other CMIP6 model outputs. Section 4 presents relevant discussions
and conclusions based on these study findings.
**2 Method**
**2.1 Chemistry–climate model**
We used the CHASER (MIROC) global chemistry–climate model (Sudo et al., 2002; Sudo and Akimoto, 2007; Watanabe et
al., 2011; Ha et al., 2021) for this study, which incorporated consideration of detailed chemical and physical processes in the
troposphere and stratosphere. The CHASER version adopted for this study simulates the distributions of 94 chemical species
while reflecting the effects of 269 chemical reactions (58 photolytic, 190 kinetic, and 21 heterogeneous). As processes
associated with tropospheric chemistry, Non-Methane Hydrocarbons (NMHC) oxidation and the fundamental chemical cycle
of O$_x$–NO$_x$–HO$_x$–CH$_4$–CO are considered. CHASER simulates stratospheric chemistry involving the Chapman mechanisms
and catalytic reactions associated with HO$_x$, NO$_x$, ClO$_x$, and BrO$_x$. Moreover, it simulates the formation of polar
stratospheric clouds (PSCs) and heterogeneous reactions occurring on their surfaces. CHASER is on-line-coupled to MIROC
AGCM ver. 5.0 (Watanabe et al., 2011), which simulates cumulus convection (Arakawa–Schubert scheme) and grid-scale
large-scale condensation to represent cloud and precipitation processes. The radiation flux is calculated using a two-stream k
distribution radiation scheme, which considers absorption, scattering, and emissions by aerosol and cloud particles as well as
by gaseous species (Sekiguchi and Nakajima, 2008; Goto et al., 2015). The aerosol component in CHASER is coupled with
the SPRINTARS aerosol model (Takemura et al., 2009), particularly for simulating primary organic carbon, sea-salt, and
dust, which is also based on MIROC. The aerosol radiation effects are considered in both large-scale condensation and
cumulus convection schemes, although the aerosol microphysical effects are only reflected in the large-scale condensation
scheme.

This study used a horizontal resolution of T42 (2.8° × 2.8°), with vertical resolution of 36 σ-p hybrid levels from the surface
to approximately 50 km. Anthropogenic and biomass burning emissions were obtained from the CMIP6 forcing datasets
(van Marle et al., 2017; Hoesly et al., 2018) for 1959–2014 (https://esgf-node.llnl.gov/search/input4mips/, last access: 19
September 2022). Interannual variation in biogenic emissions for isoprene, monoterpene, acetone, and methanol were
considered using an off-line simulation by the Vegetation Integrative Simulator for Trace Gases(VISIT)terrestrial
ecosystem model (Ito and Inatomi, 2012). The residual biogenic emissions (ethane, propane, ethylene, propene) used are
climatological values derived from the Model of Emissions of Gases and Aerosols from Nature (MEGAN) modeling system
(Guenther et al., 2012).

The CHASER (MIROC) global chemistry–climate model originally parameterizes lightning with the widely used cloud top
height scheme (Price and Rind, 1992). A newly developed ice-based lightning scheme called the ECMWF-McCAUL here
had been implemented into CHASER (MIROC) (He et al., 2022b). The ECMWF-McCAUL scheme computes LFRs as a
function of CAPE and $Q_{Ra}$ ($Q_{Ra}$ represents the total volumetric amount of cloud ice, graupel, and snow in the charge
separation region). Compared with the cloud top height, a salient advantage of the ECMWF-McCAUL scheme is that it has a
direct physical link with the charging mechanism.
**2.2 Lightning NOₓ emission parameterizations**
We tested two lightning schemes for this study. The first lightning scheme is the widely used cloud top height (CTH) scheme
(Price and Rind, 1992), which was used originally in CHASER (MIROC). This lightning scheme uses the following
equations to calculate LFR.
$F_l = 3.44 \times 10^{-5} H^{4.9}$ (1)
$F_o = 6.2 \times 10^{-4} H^{1.73}$ (2)
Therein, $F$ represents the total flash frequency (fl. min⁻¹), $H$ stands for the cloud-top height (km), and subscripts $l$ and o
respectively denote the land and ocean (Price and Rind, 1992). Actually, we realize the CTH scheme in CHASER using the
following equations (Eq. (3) and Eq. (4)) (Sudo et al., 2002). Each model layer's cumulus cloud fractions are used to weight
the calculated lightning densities from that layer in the CTH scheme.
$F_l = \sum_{i=1}^{n=36} adj\_factor \times Cu\_CF_i \times (H_i - H_{surface})^{4.9}$ (3)
$F_o = \sum_{i=1}^{n=36} adj\_factor \times Cu\_CF_i \times (H_i - H_{surface})^{1.73}$ (4)
In those equations, $i$ represents the model layer index. In addition, $adj\_factor$ represents adjustment factors that differ for
different model layers and model grids. $Cu\_CF_i$ symbolizes the cumulus cloud fraction at model layer $i$. $H_i$ and $H_{surface}$
respectively denote the altitude of model layer $i$ and the altitude of the model's surface layer.

The second lightning scheme used for this study is a newly developed one named the ECMWF-McCAUL scheme (He et al.,
2022b), which is based on the original ECMWF scheme and findings reported by McCaul et al. (2009). The ECMWF-
McCAUL scheme calculates LFRs as a function of $CAPE$ ($m^2\ s^{-2}$) and $Q_{Ra}$ ($Q_{Ra}$ symbolizes the total volumetric amount of
cloud ice, graupel, and snow in the charge separation region) as
$f_l = \alpha_l Q_{Ra} CAPE^{1.3}$ $\hspace{3cm}$ (5)
$f_o = \alpha_o Q_{Ra} CAPE^{1.3}$ $\hspace{3cm}$ (6)
where $f_l$ and $f_o$ respectively symbolize the total flash density (fl. $m^{-2}\ s^{-1}$) over land and ocean. In addition, $\alpha_l$ and $\alpha_o$ are
constants (fl. $s^{1.6}\ kg^{-1}\ m^{-2.6}$) determined after calibration against LIS/OTD climatology, respectively, for land and ocean.
For this study, $\alpha_l$ and $\alpha_o$ are set respectively as $2.67 \times 10^{-16}$ and $1.68 \times 10^{-17}$. In the charge separation region (from 0° to
-25°C isotherm), $Q_{Ra}$ ($kg\ m^{-2}$) is expressed as a proxy for the charging rate because of collisions between graupel and
hydrometeors of other types (McCaul et al., 2009). Moreover, $Q_{Ra}$ represents the total volumetric amount of hydrometeors of
three kinds (graupel, snow, and cloud ice) within the charge separation region, calculated as
$Q_{Ra} = \int_{z_0}^{z_{-25}} (q_{graup} + q_{snow} + q_{ice})\bar{\rho} dz,$ $\hspace{2cm}$ (7)
where $q_{graup}$, $q_{snow}$, and $q_{ice}$ respectively represent the mass mixing ratios ($kg\ kg^{-1}$) of graupel, snow, and cloud ice. In
addition, $q_{ice}$ was diagnosed using Arakawa–Schubert cumulus parameterization. Then, $q_{graup}$ and $q_{snow}$ were computed at
each vertical level of the model using the following equations.
$q_{graup} = \beta \dfrac{P_f}{\bar{\rho} V_{graup}}$ $\hspace{2cm}$ (8)
$q_{snow} = (1 - \beta) \dfrac{P_f}{\bar{\rho} V_{snow}}$ $\hspace{2cm}$ (9)
In those equations, $P_f$ represents the vertical profile of the frozen precipitation convective flux ($kg\ m^{-2}\ s^{-1}$), $\bar{\rho}$ denotes the
air density ($kg\ m^{-3}$), and $V_{graup}$ and $V_{snow}$ respectively express the typical fall speeds for graupel and snow set to 3.1 and 0.5
$m\ s^{-1}$ for this study. For land, the dimensionless coefficient $\beta$ is set as 0.7, whereas it is set to 0.45 for oceans to consider
the observed lower graupel content over the oceans.

Based on the cold cloud depth, a fourth-order polynomial (equation 10) is used to calculate the proportion of total flashes
that are cloud-to-ground ($p$). An earlier report of the literature describes the method (Price and Rind, 1993).
$p = \dfrac{1}{64.9 - 36.54D + 7.493D^2 - 0.648D^3 + 0.021D^4}$ $\hspace{1cm}$ (10)
The depth of the cloud above the 0°C isotherms is represented by $D$ (km) in that equation.

According to recent studies, the intra-cloud (IC) lightning flashes are as efficient as cloud-to-ground (CG) lightning flashes
at producing $NO_x$. The lightning $NO_x$ production efficiency is estimated as 100–400 mol per flash (Ridley et al., 2005;
Cooray et al., 2009; Ott et al., 2010; Allen et al., 2019). The $LNO_x$ production efficiencies for IC and CG are therefore set to
the same value (250 mol per flash) in CHASER, which is the median of the commonly cited range of 100–400 mol per flash.
Therefore, in this study, the distinctions between IC and CG do not affect the distribution or magnitude of $LNO_x$ emissions.
It is noteworthy that marked uncertainties are involved in ascertaining the $LNO_x$ production efficiency (Allen et al., 2019;
Bucsela et al., 2019). The choice of a different $LNO_x$ production efficiency might affect the simulation of $LNO_x$ emissions.
Further research must be undertaken to implement and validate a more sophisticated parameterization of $LNO_x$ production
efficiency in chemistry–climate models. The calculated total column $LNO_x$ for each grid was distributed into each model
layer based on a prescribed "backward C-shaped" $LNO_x$ vertical profile (Ott et al., 2010).
**2.3 Lightning observation data for model evaluation**
We used LIS/OTD gridded climatology datasets for this study, consisting of climatologies of total LFRs observed using the
Lightning Imaging Sensor (LIS) and Optical Transient Detector (OTD). The OTD aboard the MicroLab-1 satellite and LIS
aboard the Tropical Rainfall Measuring Mission (TRMM) satellite (Cecil et al., 2014). Both sensors detected lightning by
monitoring pulses of illumination produced by lightning in the 777.4 nm atomic oxygen multiplet above background levels.
In low Earth orbit, both sensors viewed Earth locations for approximately 3 min during the pass of the OTD or 1.5 min
during passing of the LIS. Each day, OTD and LIS respectively orbited the globe 14 times and 16 times. OTD observed data
between +75 and -75° latitude during May 1995 – March 2000, whereas LIS collected data between +38 and -38° latitude
during January 1998 – April 2015. This study uses the LIS/OTD 2.5 Degree Low Resolution Time Series (LRTS), which
provides daily LFRs on a 2.5° regular latitude–longitude grid for May 1995 – April 2015.
**2.4 CMIP6 model outputs for model comparison**
For the comparison of different model outputs from our study (CHASER) and other Earth system models or chemistry–
climate models, we used LFR and surface temperature data from the CMIP6 CMIP Historical experiments from CESM2-
WACCM (Danabasoglu, 2019), GISS-E2-1-G (Kelley et al., 2020), and UKESM1-0-LL (Tang et al., 2019). CESM2-
WACCM uses the Community Earth System Model ver. 2 (Danabasoglu et al., 2020). The CESM2 is an open-source fully
coupled Earth system model. The Whole Atmosphere Community Climate Model ver. 6 (WACCM6) is the atmospheric
component coupled to the other components in CESM2. The GISS-E2-1-G is the NASA Goddard Institute for Space Studies
(GISS) chemistry–climate model version E2.1 based on the GISS Ocean v1 (G01) model (Miller et al., 2014; Kelley et al.,
2020). The UKESM1-0-LL is the UK's Earth system model, details of which were described by Sellar et al. (2019). We used
3 ensembles from CESM2-WACCM, 9 ensembles from GISS-E2-1-G, and 18 ensembles from UKESM1-0-LL. Table S1
presents all the ensemble members used for this study.

## 2.5 Experiment setup

We have conducted six sets of experiments with each set of experiments conducted using both the ECMWF-McCAUL (abbreviated as F1) and CTH (abbreviated as F2) schemes. Table 1 presents the major settings of all experiments with the relative explanations of those settings. STD-F1/F2 are standard experiments with the simulation period of 1959–2014. They are intended to reproduce the historical trends of lightning and LNO$_x$. Climate1959-F1/F2 are experiments that keep the climate simulations fixed to 1959 to derive the effects of global warming on historical lightning trends. ClimateAero1959-F1/F2 are intended to reflect the conditions with climate simulations and aerosol and aerosol precursor (BC, OC, NO$_x$, SO$_2$) emissions fixed to 1959. The Aero1959-F1/F2 experiments are the same as the STD-F1/F2 experiments, except for the AeroPEs fixed to 1959. The fifth set of experiments (Volca-off-F1/F2) was intended to exclude the influences of the Pinatubo volcanic eruption to compare to the STD-F1/F2 and to evaluate the Pinatubo eruption effects on historical lightning–LNO$_x$ trends and variation.

We simulate volcanic aerosol forcing by considering the prescribed stratospheric aerosol extinction in the radiation scheme. We used the NASA Goddard Institute for Space Studies (GISS) (Sato et al., 1993) and Chemistry–Climate Model Initiative (CCMI) (Arfeuille et al., 2013) stratospheric aerosol dataset as the stratospheric aerosol climate data. The NASA GISS dataset includes monthly zonal-mean stratospheric aerosol optical thickness in four spectral bands. The CCMI dataset for CHASER includes monthly zonal-mean stratospheric aerosol extinction coefficients in 20 spectral bands. To remove the volcanic perturbation while maintaining the stratospheric background aerosol in the Volca-off-F1/F2, we used the following equation to process the Stratospheric Aerosol Climatology (SAC) during June 1991 – May 1996.

$$SAC_{no\_pinatubo} = \begin{cases} SAC_{background}, |SAC_{raw} - SAC_{background}| > 1.96\sigma, \\ SAC_{raw}, |SAC_{raw} - SAC_{background}| \leq 1.96\sigma \end{cases} \quad (11)$$

In that equation, $SAC_{no\_pinatubo}$ denotes the stratospheric aerosol climatological data as input data for Volca-off-F1/F2 experiments, $SAC_{background}$ represents the stratospheric background aerosol climatological data (For this study, $SAC_{background}$ is the corresponding temporal averaged values of the NASA GISS and CCMI stratospheric aerosol dataset during June 1986 – May 1991 and June 1996 – May 2001, when the time is close to the eruption and the stratosphere was less affected by volcanic eruptions). $SAC_{raw}$ stands for the original values of NASA GISS and CCMI stratospheric aerosol dataset during June 1991 – May 1996. Moreover, $\sigma$ symbolizes the standard deviations of stratospheric background aerosol climate data (For this study, $\sigma$ are the corresponding standard deviations of NASA GISS and CCMI stratospheric aerosol dataset during June 1986 – May 1991 and June 1996 – May 2001). As displayed in Eq. (11), when the absolute differences between $SAC_{raw}$ and $SAC_{background}$ are larger than 1.96$\sigma$, we replace the original values (June 1991 – May 1996) of the SAC with the temporal averaged values of the NASA GISS and CCMI dataset during June 1986 – May 1991 and June 1996 – May 2001. When the absolute differences between $SAC_{raw}$ and

$SAC_{background}$ are equal to or smaller than $1.96\sigma$, we still use the original values (June 1991 – May 1996) of the SAC
for the Volca-off experiments. The value of $1.96\sigma$ corresponds to the 95% confidence interval, which can remove the
Pinatubo perturbation sufficiently but which can maintain the background level of stratospheric aerosol during June
1991 – May 1996. Furthermore, the influences of the Pinatubo eruption affected the HadISST SSTs/sea ice fields. To
remove the Pinatubo eruption's influences on the SSTs/sea ice fields from the Volca-off experiments also, we replaced
the 1991-06 – 1995-05 SSTs/sea ice data with HadISST SSTs/sea ice climatological data during 1985–1990 when
conducting the Volca-off experiments. The 1985–1990 period was chosen because it is approximately the period of
1991-06 – 1995-05 and because the SSTs/sea ice fields were less affected by volcanic activity during 1985–1990.

All the experiments calculate the $LNO_x$ emissions rates interactively by $LNO_x$ emission parameterizations except STD-
rVolcaoff experiments. The STD-rVolcaoff experiments are the same as the STD experiments except for reading the
daily $LNO_x$ emission rates calculated from the Volca-off experiments. The STD-rVolcaoff experiments are conducted
for comparison with STD experiments to elucidate the effects of $LNO_x$ emissions changes caused by the Pinatubo
eruption on atmospheric chemistry (typically methane lifetime).
**Table 1: All experiments conducted for this study**

| Name of experiment | Period | Climate (SSTs, sea ice, GHGs)[a] | Anthropogenic and biomass burning emissions | Biogenic emissions | Stratospheric aerosol climatology |
|---|---|---|---|---|---|
| STD-F1/F2[b] | 1959–2014 | 1959–2014 | CMIP6 1959–2014 | | NASA GISS and CCMI stratospheric aerosol dataset[c] |
| Climate1959-F1/F2 | 1959–2014 | Fixed to 1959[d] | CMIP6 1959–2014 | VISIT and MEGAN[f] | As above |
| ClimateAero1959-F1/F2 | 1959–2014 | Fixed to 1959 | AeroPEs fixed to 1959[e] | | As above |
| Aero1959-F1/F2 | 1959–2014 | 1959–2014 | AeroPEs fixed to 1959 | | As above |
| Volca-off-F1/F2 | 1990–1999 | 1990–1999[g] | CMIP6 1990–1999 | | Same dataset with volcanic perturbation removed |
| STD-rVolcaoff-F1/F2 | 1990–1999 | All settings are the same as those used for STD experiment except for reading of the daily $LNO_x$ emission rates calculated from the Volca-off experiments | | | |

[a] For the model simulations, the climate is simulated by the prescribed SSTs/sea ice fields and the prescribed varying
concentrations of GHGs ($CO_2$, $N_2O$, methane, chlorofluorocarbons – CFCs – and hydrochlorofluorocarbons – HCFCs) used
only in the radiation scheme. The SSTs/sea ice fields are obtained from the HadISST dataset (Rayner et al., 2003). The
prescribed GHGs concentrations are derived from CMIP6 forcing datasets (Meinshausen et al., 2017).
[b] We use "F1" to stand for the ECMWF-McCAUL scheme; "F2" represents the CTH scheme.
[c] Stratospheric aerosol radiative forcing is simulated using the prescribed stratospheric aerosol extinction, which is obtained
from the NASA GISS (Sato et al., 1993) and CCMI (Arfeuille et al., 2013) stratospheric aerosol dataset.
[d] The climate is fixed to 1959 for the whole simulation period using the 1959 SSTs/sea ice field and GHG concentrations
during the simulation period.
[e] Aerosol (BC, OC) and aerosol precursor ($NO_x$, $SO_2$) emissions (anthropogenic + biomass burning) are fixed to 1959
throughout the simulation period.
[f] Several biogenic emissions are interannually varying, including isoprene, monoterpenes, acetone, and methanol, which
were calculated using an off-line simulation using the Vegetation Integrative Simulator for Trace Gases (VISIT) terrestrial
ecosystem model (Ito and Inatomi, 2012). Some other reactive biogenic VOCs (ethane, propane, ethylene, propene) used are
climatological data derived from the Model of Emissions of Gases and Aerosols from Nature (MEGAN) modeling system
(Guenther et al., 2012).
[g] Here the 1991-06 – 1995-05 SSTs/sea ice data were replaced with HadISST SSTs/sea ice climatological data during
1985–1990.
**3 Results and Discussion**
**3.1 Validation of the simulated historical lightning distribution and trend**
To increase the credibility of the conclusions obtained based only on the numerical simulations, the model calculations must
be evaluated using observational data. We used the LIS/OTD observations to evaluate the spatial and temporal distribution
and historical lightning trends simulated by CHASER (MIROC). Figures 1a–1c show the annual mean spatial distributions
of lightning observed by LIS/OTD and from model simulations using the ECMWF-McCAUL and CTH schemes. Both the
ECMWF-McCAUL and CTH schemes generally captured the hotspots of lightning (Central Africa, Maritime Continent,
South America), with strong spatial correlations between observations and model simulations ($R > 0.75$), even the lightning
distributions were not well captured over the ocean. Figure 1d exhibits strong spatial correlation between observations and
simulation results maintained throughout the simulation period (1959–2014).

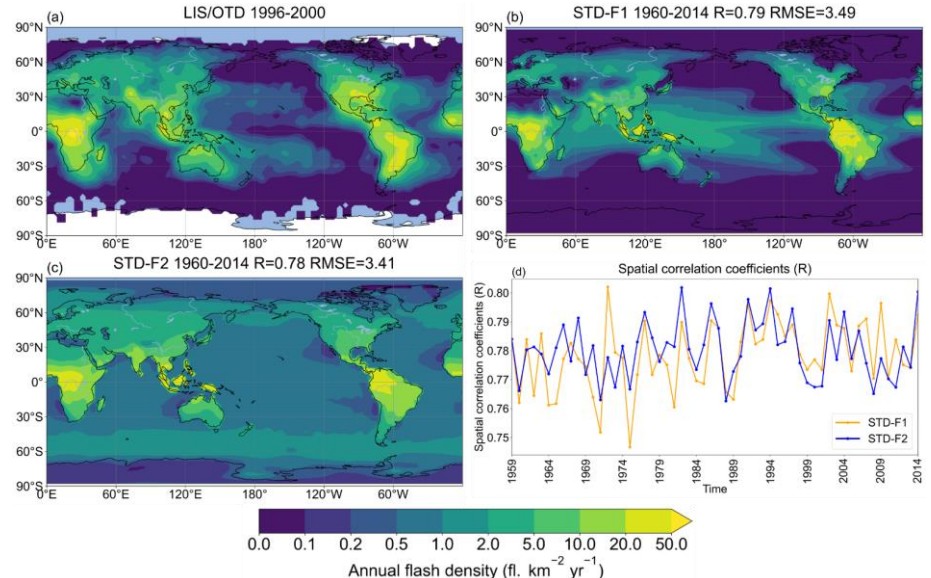


**Figure 1: Annual mean lightning flash densities from (a) LIS/OTD satellite observations spanning 1996–2000, (b) the STD**
**experiment (1960–2014) with the ECMWF-McCAUL scheme used, (c) the STD experiment (1960–2014) with the CTH scheme**
**used. *R* and RMSE shown in the titles of panels (b) and (c) are calculated between panels (b)–(c) and (a). Panel (d) presents the**
**spatial correlation coefficients between modeled spatial lightning distribution of each year and LIS/OTD lightning climatologies**
**during 1996–2000.**

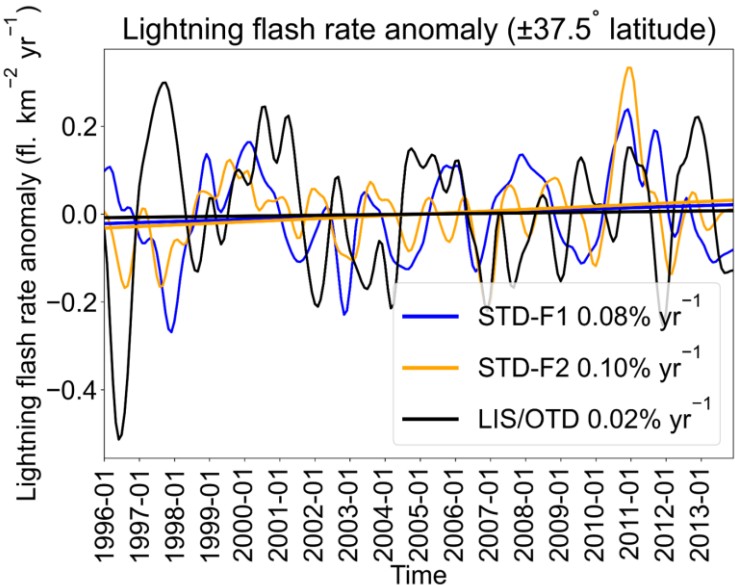


**Figure 2: LFR anomalies of 1996–2013 within ±37.5° latitude obtained from two numerical experiments (STD-F1/F2) and**
**LIS/OTD satellite observations. Curves represent the monthly time-series data of the ±37.5° latitude mean LFR anomalies with the**
**1-D Gaussian (Denoising) filter applied. Lines are the fitting curves of the monthly time-series data of the ±37.5° latitude mean**
**LFR anomalies. Trends of the LFR anomalies in % yr$^{-1}$ are also presented in the legends.**

**Table 2: A statistical summary of the trends shown in Fig. 2 by Mann–Kendall rank statistic and Sen's slope estimator. The**
**monthly time-series data of the ±37.5° latitude mean LFR anomalies were estimated by Mann–Kendall rank statistic and Sen's**
**slope estimator. The column "Trend" shows whether these are significant trends with the significance set as 5%, as well as the**
**percentage trends in % yr$^{-1}$ estimated by linear regression. The "$p$-value" is calculated during Mann-Kendall trend test. "Slope"**
**shows Sen's slope of trend. $Q_{min}$ and $Q_{max}$ respectively denote the lower and upper limits of the 95% confidence interval of Sen's**
**slope.**

| Experiment/dataset | Trend | $p$-value | Slope | $Q_{min}$ | $Q_{max}$ |
|---|---|---|---|---|---|
| STD-F1 | No trend, 0.08 % yr$^{-1}$ | $p > 0.05$ | 0.0001 | -0.0003 | 0.0005 |
| STD-F2 | No trend, 0.10 % yr$^{-1}$ | $p > 0.05$ | 0.0003 | 0.0 | 0.0006 |
| LIS/OTD | No trend, 0.02 % yr$^{-1}$ | $p > 0.05$ | -0.0001 | -0.0006 | 0.0004 |


The LIS/OTD observations are also used to evaluate historical lightning trends simulated by CHASER (MIROC). We
examined the ±37.5° latitude mean LFR anomaly (1996–2013) calculated from LIS/OTD observations and STD-F1/F2
numerical experiments (Fig. 2 and Table 2). We also note some missing values within the ±37.5° latitude in LIS/OTD
observations. To constrain the comparisons between observations and simulations as like-for-like, when we encounter a
missing value in the LIS/OTD observations during spatial averaging, we also treat the CHASER simulated value at the same
location as a missing value. As displayed in Fig. 2, we would not necessarily expect that interannual variations of LFR
anomaly can be captured, because meteorological nudging was not applied and the simulated LFRs were only controlled by
the prescribed SSTs/sea ice data. Nevertheless, the overall trends of LFR anomaly simulated using both schemes well-
matched the LIS/OTD observations, as portrayed in Fig. 2. We further investigated the trends shown in Fig. 2 by Mann–
Kendall rank statistic and Sen's slope estimator and the statistical summary is displayed in Table 2 (T. et al., 2002; Hussain
and Mahmud, 2019). Neither the LFR anomaly (within ±37.5° latitude) derived from LIS/OTD observations nor simulations
show a significant trend for 1996–2013 using the Mann–Kendall rank statistic test (significance inferred for 5%). The global
LFR anomaly during 1993–2013 obtained from simulations (STD-F1/F2) also shows no significant trend, which is consistent
with the Schuman Resonance (SR) intensity observations (1993–2013) at Rhode Island, USA (Earle Williams, 2022).
However, the SR observations in Rhode Island (USA) exclude consideration of the influences of solar cycles, which makes it
less appropriate for lightning trend evaluation.

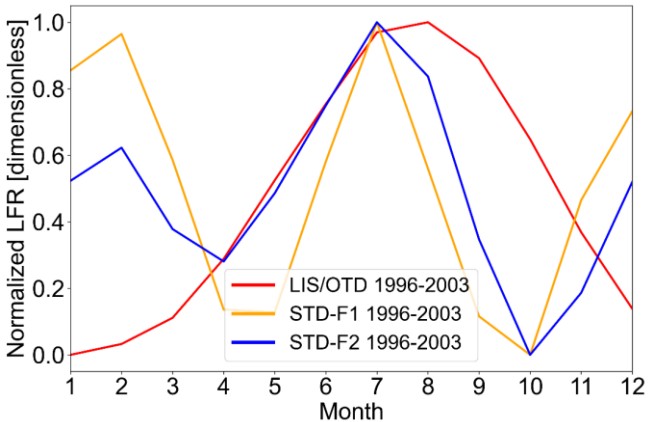

**Figure 3: Mean annual cycle in area average LFR during 1996–2003. The area average was taken over the grid cells where valid LIS/OTD lightning observations exist. LFR is normalized by min-max normalization.**

We further investigated the seasonal variabilities of simulated LFR and compared them against LIS/OTD observations. The results are depicted in Fig. 3. Both the CTH and ECMWF-McCAUL schemes captured the peak during JJA, but the overestimation of LFR by F1/F2 during DJF is also noticeable. Figure S1 presents comparison of the LFR global distribution in different seasons during 1996–2003 from LIS/OTD lightning observations and STD experiment outputs. Generally, CHASER well-captured the spatial distribution of LFR in all four seasons when compared against LIS/OTD observations. The spatial correlation coefficients ($R$) between observations and simulations are highest ($R$=0.80 for both lightning schemes) in DJF, indicating CHASER's considerable capability to reproduce the LFR spatial distribution in DJF. As displayed in the first row of Fig. S1, the overestimation of LFR by F1/F2 during DJF is primarily attributable to the overestimation of LFR within the Maritime Continent and South America, but this might also be attributable to the underestimation of LFR by LIS/OTD within these two regions. It is believed that the LIS/OTD lightning detection efficiency is highly sensitive to the characteristic of convective clouds (cloud albedo, cloud optical thickness, etc.) (Boccippio et al., 2002; Cecil et al., 2014). High cloud albedo and cloud optical thickness might engender the underestimation of LFR by LIS/OTD. It is also noteworthy that the seasonal variation and long-term trend of global lightning are strongly influenced by distinct different factors. The seasonal variation of global lightning activities is most strongly affected by the 23° obliquity of Earth's orbit and the asymmetric distribution of the continent between the Northern and Southern hemispheres. However, the long-term global lightning trend we investigated for this study is controlled mainly by climate forcers such as aerosols and GHGs. To minimize the effects of LFR seasonal variation on our study's results, we deseasonalized the results shown in all figures and tables by calculating their anomaly based on raw data. The validation described above and the deseasonalization of our study's results justified that the LFR seasonal variation (and the uncertainties in the simulation of LFR seasonal variation) in our study has a limited effect on these study results.

## 3.2 Effects of global warming and increases in AeroPEs on historical lightning–LNO$_x$ trends

As introduced in Sect. 1, global warming and changes in AeroPEs are the two main factors which influence long-term (1960–2014) historical lightning trends (Hereinafter, historical lightning trends represent lightning trends of 1960–2014.). Evidence shows that the Pacific Decadal Oscillation (PDO) can also affect lightning trends over decadal time scales (Macias Fauria and Johnson, 2006; Mallick et al., 2022), and further research is anticipated to verify it. To analyze the effects of global warming on historical lightning trends, we designed and conducted two sets of experiments: one set of experiments including "global warming" (STD-F1/F2) and another set of experiments excluding "global warming" (Climate1959-F1/F2). Figures 4a and 4b respectively depict the global surface temperature anomalies calculated using the ECMWF-McCAUL and CTH schemes. The STD and Aero1959 experiments show an increasing trend (around 0.11 K decade$^{-1}$) of global mean surface temperature anomalies, which closely approximates the trend (around 0.15 K decade$^{-1}$) obtained from NOAA's National Centers for Environmental Information (NCEI) (Figs. 4c, 4f). Global temperature change data from 1880 to the present are available from the NCEI, which tracks variations of the Earth's temperature based on thousands of stations' observation data around the globe (Climate at a Glance | National Centers for Environmental Information (NCEI), 2022). When the prescribed SSTs/sea ice fields and GHGs concentrations were fixed to 1959 throughout the simulation period, the simulated trends of global mean surface temperature anomalies turned out to be flat (Climate1959 and ClimateAero1959). To elucidate the effects of increases in AeroPEs on averaged surface temperature to the greatest extent possible, we also show the averaged surface temperature anomaly only over land regions (Figs. 4d–4f). The simulated global mean land surface temperature anomalies are also well-matched with the NCEI observational data. The aerosol cooling effect can be more evident when only examining surface temperature trends averaged over land (Figs. 4d–4e).

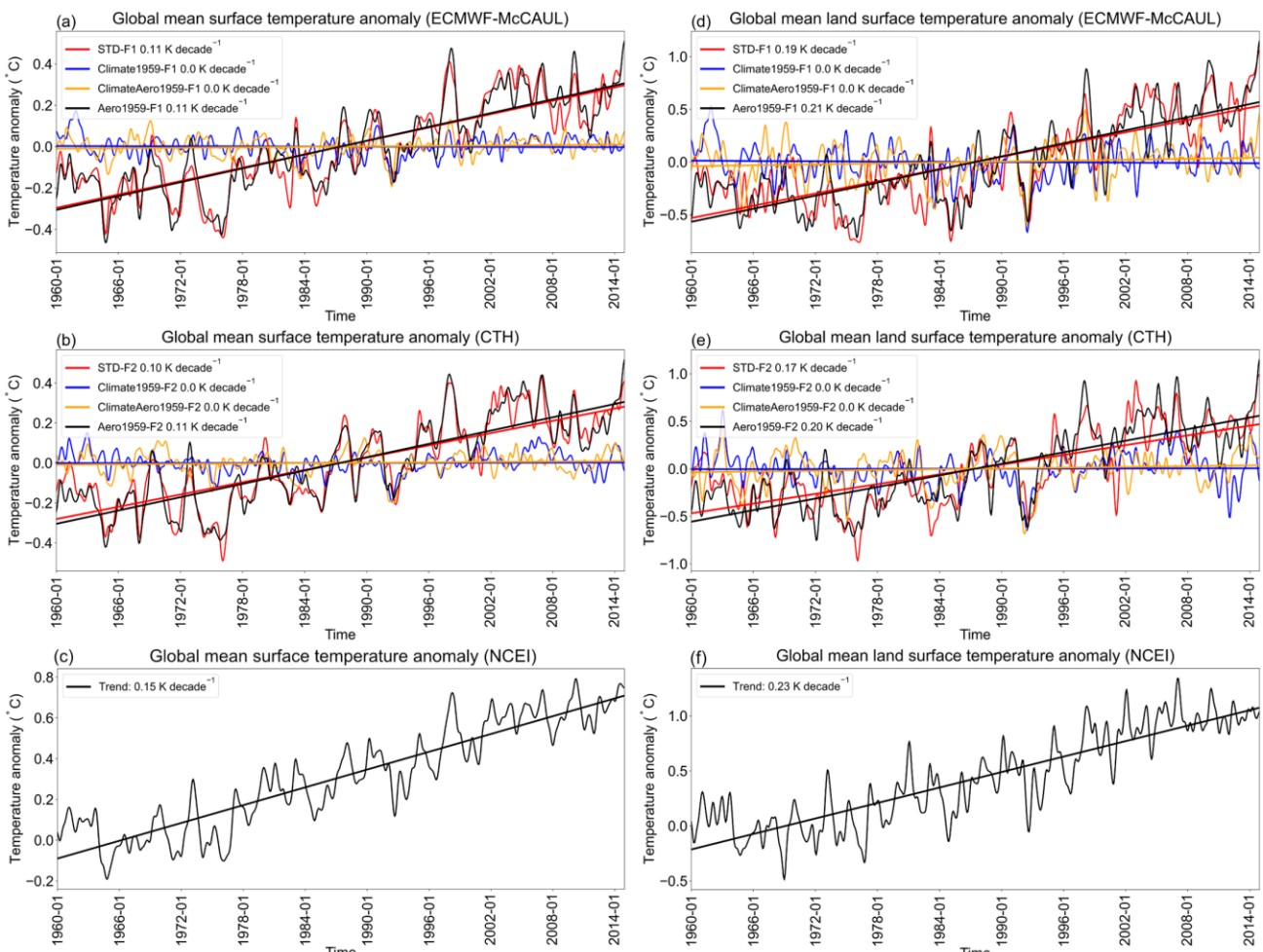

373

**Figure 4: Monthly time-series data of global mean surface temperature anomalies with 1-D Gaussian (Denoising) filter applied and their fitting curves calculated from the outputs of numerical experiments (a–b) and obtained from NCEI (c). Panels (d)–(f) are the same as panels (a)–(c), but the averaged surface temperature anomalies are only calculated within the global land regions. The trends of the fitting curves in K decade$^{-1}$ are also presented in the legends.**

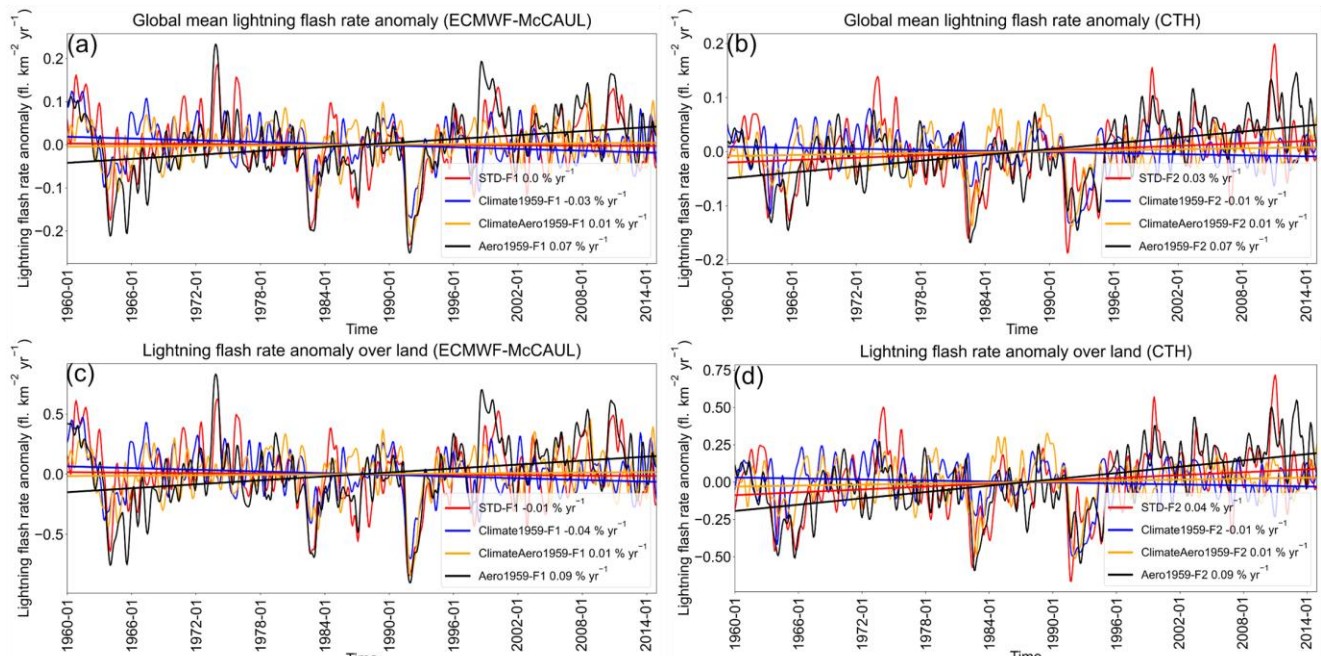


**Figure 5: Panels (a) and (b) show monthly time-series data of global mean LFR anomalies with 1-D Gaussian (Denoising) Filter**
**applied and their fitting curves of different experiments simulated respectively using the ECMWF-McCAUL scheme and CTH**
**scheme. Panels (c) and (d) are the same as panels (a) and (b), except that the averaged LFR anomalies are calculated only within**
**global land regions. Trends of the fitting curves (% yr$^{-1}$) are also shown in the legends.**

**Table 3: A statistical summary of the trends shown in Fig. 5 by Mann–Kendall rank statistic and Sen's slope estimator. The**
**monthly time-series data of global or land mean LFR anomalies were estimated by Mann–Kendall rank statistic and Sen's slope**
**estimator. The column "Trend" shows whether these are significant trends with the significance set as 5%, as well as the**
**percentage trends in % yr$^{-1}$ estimated by linear regression. The "$p$-value" is calculated during Mann-Kendall trend test. "Slope"**
**shows Sen's slope of trend. $Q_{min}$ and $Q_{max}$ respectively denote the lower and upper limits of the 95% confidence interval of Sen's**
**slope.**

| Experiment | Trend | $p$-value | Slope | $Q_{min}$ | $Q_{max}$ |
|---|---|---|---|---|---|
| STD-F1 (global) | No trend, 0.0 % yr$^{-1}$ | $p > 0.05$ | 0.0 | -0.0001 | 0.0 |
| Climate1959-F1 (global) | Decreasing, -0.03 % yr$^{-1}$ | $p < 0.01$ | -0.0001 | -0.0001 | 0.0 |
| ClimateAero1959-F1 (global) | No trend, 0.01 % yr$^{-1}$ | $p > 0.05$ | 0.0 | 0.0 | 0.0001 |
| Aero1959-F1 (global) | Increasing, 0.07 % yr$^{-1}$ | $p < 0.01$ | 0.0001 | 0.0001 | 0.0002 |
| STD-F1 – Climate1959-F1 (global) | No trend, 0.03 % yr$^{-1}$ | $p > 0.05$ | 0.0001 | 0.0 | 0.0001 |
| STD-F1 – Aero1959-F1 (global) | Decreasing, -0.07 % yr$^{-1}$ | $p < 0.01$ | -0.0001 | -0.0002 | -0.0001 |
| STD-F1 (land) | No trend, -0.01 % yr$^{-1}$ | $p > 0.05$ | 0.0 | -0.0002 | 0.0001 |
| Climate1959-F1 (land) | Decreasing, -0.04 % yr$^{-1}$ | $p < 0.01$ | -0.0002 | -0.0004 | -0.0001 |

| | | | | | |
|---|---|---|---|---|---|
| ClimateAero1959-F1 (land) | No trend, 0.01 % yr$^{-1}$ | $p > 0.05$ | 0.0001 | -0.0001 | 0.0002 |
| Aero1959-F1 (land) | Increasing, 0.09 % yr$^{-1}$ | $p < 0.01$ | 0.0005 | 0.0003 | 0.0006 |
| STD-F1 – Climate1959-F1 (land) | No trend, 0.03 % yr$^{-1}$ | $p > 0.05$ | 0.0002 | -0.0001 | 0.0004 |
| STD-F1 – Aero1959-F1 (land) | Decreasing, -0.10 % yr$^{-1}$ | $p < 0.01$ | -0.0005 | -0.0007 | -0.0003 |
| STD-F2 (global) | Increasing, 0.03 % yr$^{-1}$ | $p < 0.01$ | 0.0001 | 0.0 | 0.0001 |
| Climate1959-F2 (global) | No trend, -0.01 % yr$^{-1}$ | $p > 0.05$ | 0.0 | -0.0001 | 0.0 |
| ClimateAero1959-F2 (global) | No trend, 0.01 % yr$^{-1}$ | $p > 0.05$ | 0.0 | 0.0 | 0.0001 |
| Aero1959-F2 (global) | Increasing, 0.07 % yr$^{-1}$ | $p < 0.01$ | 0.0001 | 0.0001 | 0.0002 |
| STD-F2 – Climate1959-F2 (global) | Increasing, 0.04 % yr$^{-1}$ | $p < 0.01$ | 0.0001 | 0.0 | 0.0001 |
| STD-F2 – Aero1959-F2 (global) | Decreasing, -0.04 % yr$^{-1}$ | $p < 0.01$ | -0.0001 | -0.0001 | 0.0 |
| STD-F2 (land) | Increasing, 0.04 % yr$^{-1}$ | $p < 0.01$ | 0.0003 | 0.0001 | 0.0004 |
| Climate1959-F2 (land) | No trend, -0.01 % yr$^{-1}$ | $p > 0.05$ | -0.0001 | -0.0002 | 0.0 |
| ClimateAero1959-F2 (land) | No trend, 0.01 % yr$^{-1}$ | $p > 0.05$ | 0.0001 | 0.0 | 0.0002 |
| Aero1959-F2 (land) | Increasing, 0.09 % yr$^{-1}$ | $p < 0.01$ | 0.0006 | 0.0004 | 0.0007 |
| STD-F2 – Climate1959-F2 (land) | Increasing, 0.05 % yr$^{-1}$ | $p < 0.01$ | 0.0003 | 0.0001 | 0.0005 |
| STD-F2 – Aero1959-F2 (land) | Decreasing, -0.05 % yr$^{-1}$ | $p < 0.01$ | -0.0003 | -0.0005 | -0.0001 |


Figure 5, panels (a) and (b) respectively portray the global mean LFR anomalies and their fitting curves obtained from the
outputs of the ECMWF-McCAUL scheme and CTH scheme. Besides, we displayed in Table 3 the statistical summary of the
trends in Fig. 5 utilizing Mann–Kendall rank statistic and Sen's slope estimator. The global lightning trend obtained from the
STD-F1 experiment turned out to be statistically flat (0.0% yr$^{-1}$), whereas the outputs of the STD-F2 experiment exhibit a
significant increasing global lightning trend (0.03% yr$^{-1}$) determined using the Mann–Kendall rank statistic (significance
inferred for 5%).

Comparison of the lightning trends calculated from the STD and Climate1959 experiments showed that both lightning
schemes demonstrated that historical global warming (1960–2014) enhances the global lightning trends toward positive
trends (around 0.03% yr$^{-1}$ or 3% K$^{-1}$). Global warming effects on historical lightning trends were evaluated as significant
using the Mann–Kendall rank statistic, with significance inferred for 5%, when using the CTH scheme, but not in the case of
the ECMWF-McCAUL scheme (see rows "STD-F1 – Climate1959-F1 (global)" and "STD-F2 – Climate1959-F2 (global)
" in Table 3). As shown in Table 3, the differences in global lightning trends simulated by the STD-F1/F2 and Aero1959-
F1/F2 experiments indicate that the increases in AeroPEs during 1960–2014 significantly suppress the global lightning
trends (-0.07% yr$^{-1}$ – -0.04% yr$^{-1}$). It is noteworthy that this suppression of lightning trends is only attributable to aerosol
radiative effects. Further research must be conducted to elucidate the long-term effects of aerosols on lightning through
aerosol microphysical effects. We also investigated lightning trends only over land regions (Figs. 5c–5d and Table 3) to
ascertain the effects of changes in AeroPEs to the greatest extent possible. When observing the lightning trends over land
only, the degree of suppression of lightning trends attributable to increases in AeroPEs expands to -0.10% yr$^{-1}$ – -0.05% yr$^{-1}$,
which is attributable to most AeroPEs and their growth coming from land regions. It is noteworthy that we used the same
SSTs/sea ice data in the Aero1959 as those used for STD experiments. The SSTs/sea ice data also reflected the effects of
increases in AeroPEs. Therefore, we might underestimate the effects of increases in AeroPEs on lightning trends by
comparing the results of STD and Aero1959 experiments.

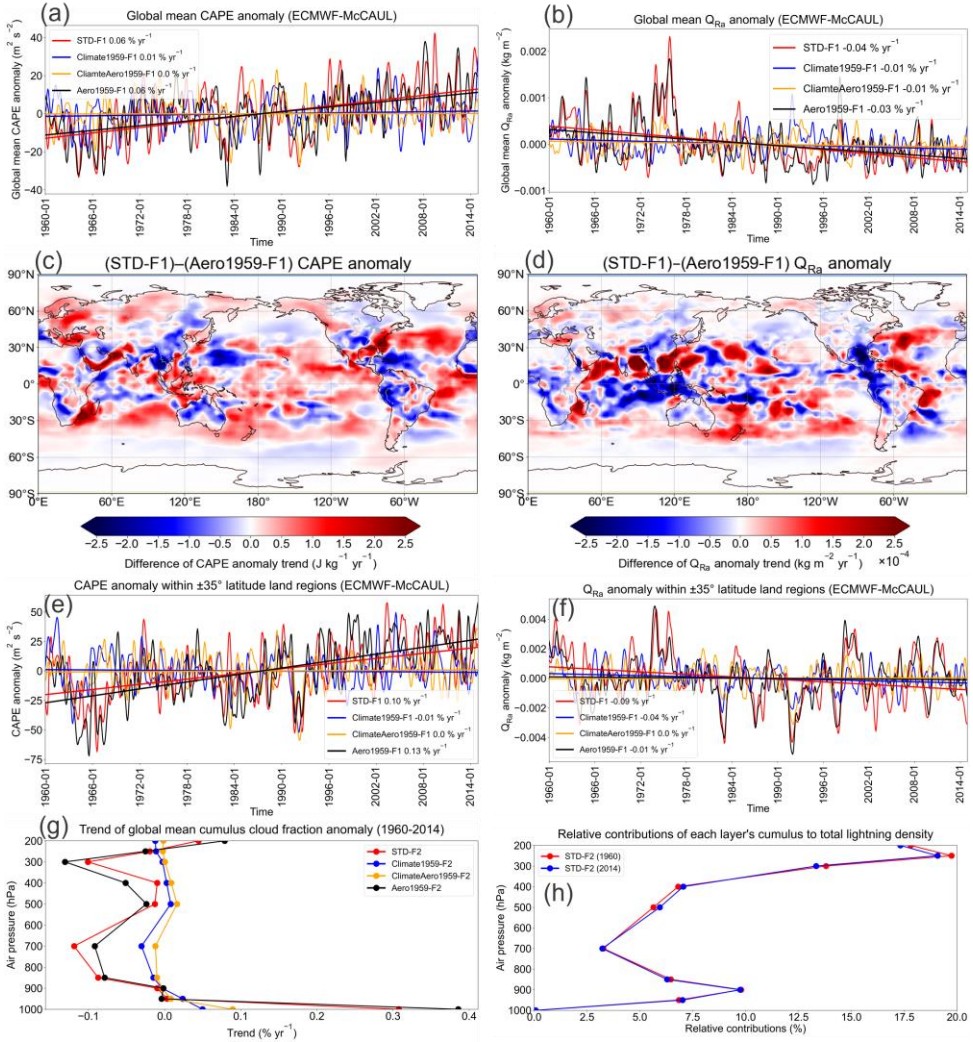


**Figure 6: Panels (a) and (b) respectively show monthly time-series data of global mean CAPE and $Q_{Ra}$ anomalies with 1-D**
**Gaussian (Denoising) filter applied and their fitting curves simulated using the ECMWF-McCAUL scheme. Panels (c) and (d)**
**respectively show differences in the CAPE anomaly trend $(\mathrm{J\,kg^{-1}\,yr^{-1}})$ and $Q_{Ra}$ anomaly trend $(\mathrm{kg\,m^{-2}\,yr^{-1}})$ of the STD-F1 and**
**Aero1959-F1 experiments in the global map. Panels (e) and (f) respectively show monthly time-series data of $\pm 35°$ latitude land**
**region mean CAPE and $Q_{Ra}$ anomalies with 1-D Gaussian (Denoising) filter applied and their fitting curves simulated using the**

**ECMWF-McCAUL scheme. Figure 6(g) portrays the vertical profiles of the trend of global mean cumulus cloud fraction anomaly simulated by the CTH scheme. Panel (h) depicts the relative contributions of each layer's cumulus to total lightning density in 1960 and 2014, as calculated from the outputs of the STD-F2 experiment.**

For the ECMWF-McCAUL scheme, model outputs affirm that global warming can enhance the global mean CAPE anomaly slightly and suppress the global mean $Q_{Ra}$ anomaly (Figs. 6a–6b). Earlier studies have also indicated that the total solid (cloud ice, snow, and graupel) mass mixing ratio within charge separation regions is lower under global warming. Moreover, possible explanations are given in those studies (Finney et al., 2018; Romps, 2019). Because global warming enhances global convection activities, and because lightning formation is highly related to convection activity, global warming enhances the historical global lightning trend simulated using the ECMWF-McCAUL scheme, mainly as a result of the simulated CAPE trend, which is enhanced by global warming.

The past increases in AeroPEs exert negligible effects on the trends of global mean CAPE and $Q_{Ra}$ anomalies, as displayed in Figs. 6a–6b. However, as also demonstrated in our study (see Fig. 1), most lightning flashes occur over tropical and subtropical land regions. It is displayed in Figs. 6c–6d that the past increases in AeroPEs mostly suppress the CAPE and $Q_{Ra}$ absolute trends within regions with high lightning densities. We further investigated the trends of $\pm 35°$ latitude land region mean CAPE and $Q_{Ra}$ anomalies, and the results are portrayed in Figs. 6e–6f. Figs. 6e–6f show that past increases in AeroPEs significantly suppress the $Q_{Ra}$ trend (-0.08 % yr$^{-1}$) and slightly suppress the CAPE trend (-0.03 % yr$^{-1}$) within $\pm 35°$ latitude land regions. Weaker convection activities (smaller CAPE) and fewer hydrometeors (cloud ice, graupel, snow) in the charge separation regions (0°C – -25°C isotherm) engender less lightning. In the case of the ECMWF-McCAUL scheme, CAPE and $Q_{Ra}$ trends were suppressed within $\pm 35°$ latitude terrestrial regions. This constitutes the main reason for the suppression of the historical global lightning trends induced by increases in AeroPEs through aerosol radiative effects. It is noteworthy that, because the aerosol microphysical effects are only considered in the grid-scale large-scale condensation scheme, our study might underestimate the aerosol microphysical effects which can enhance the trends of $Q_{Ra}$ and LFR toward the positive direction.

To explain the results simulated by the CTH scheme, we investigated the vertical profiles of the trend of the global mean cumulus cloud fraction anomaly (Fig. 6g). Investigating cumulus cloud fraction is reasonable because each model layer's cumulus cloud fractions are used to weight the calculated lightning densities from that layer in the CTH scheme, as introduced in equations (3) and (4). Figure 6h shows the relative contributions of each model layer's cumulus to the calculated global total lightning densities in 1960 and 2014 obtained using the CTH scheme. As Fig. 6h displayed, the vertical profiles of relative contribution in 1960 and 2014 are almost identical. Cumulus convection is positively correlated with lightning formation, which is the scientific basis of parameterizing lightning densities using the cumulus cloud top height: the CTH scheme. Historical global warming enhances the lightning trend simulated by the CTH scheme mainly

because the simulated historical global warming increases the cumulus reaching 200 hPa, which contributes greatly to the
simulated global total lightning density (Figs. 6g–6h). The increases in the deep convective cloud are regarded as related to
the increases in tropopause height attributable to global warming, as shown in Fig. S2. The past increases in AeroPEs
suppress the lightning trend simulated by the CTH scheme because increases in AeroPEs decrease the cumulus reaching 200
hPa as well as the cumulus within the lower to middle troposphere by aerosol radiative effects (Fig. 6g). In addition, in the
supplement, we present a figure (Fig. S3) resembling Fig. 6, but which includes only consideration of land regions. The
mechanisms of global warming and increases in AeroPEs affecting lightning trends over land regions are similar to those
described above on a global scale. We do not discuss details of them here.

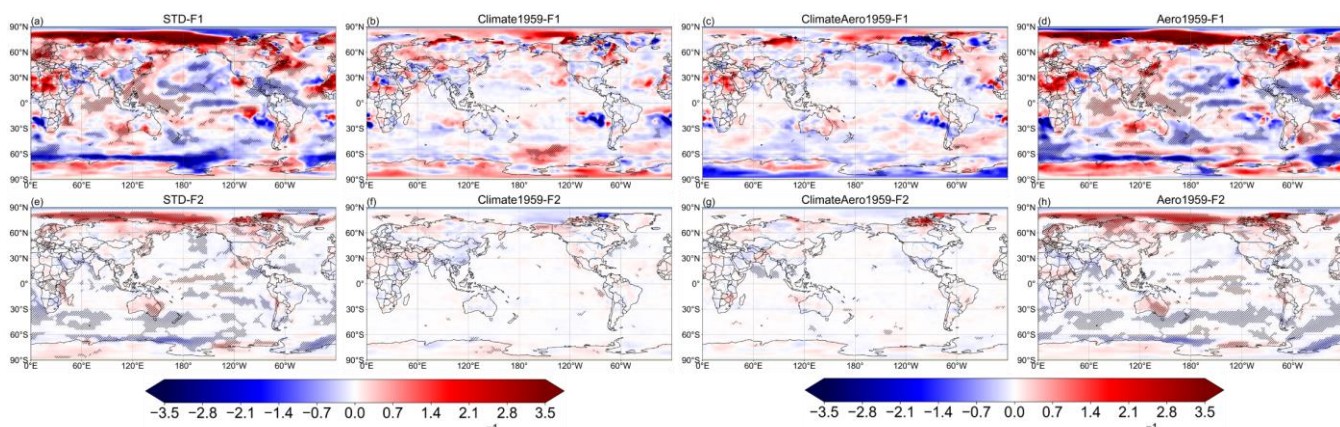


**Figure 7: Trends of LFR anomaly (% yr[-1]) during 1960–2014 on the two-dimensional map. The trend at every point was calculated**
**from the function of approximating curve for the 1960–2014 time-series data (LFR anomaly) at each grid cell. The area in which**
**the trend was found to be significant by the Mann–Kendall rank statistic test (significance inferred for 5%) is marked with**
**hatched lines.**

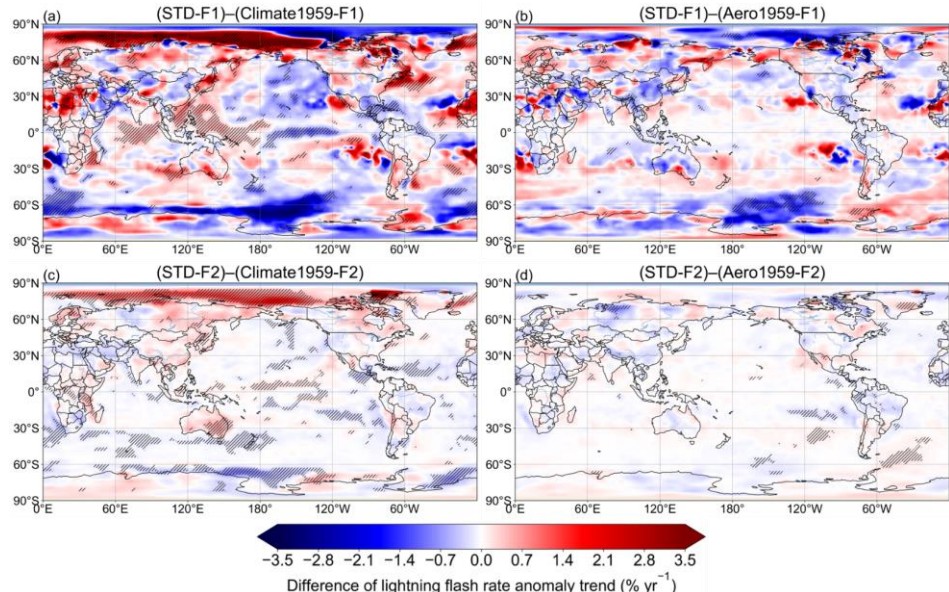


**Figure 8: Differences in trends of LFR anomaly during 1960–2014 on the global map. The area in which the trend of the differences of LFR anomaly time-series data was found to be significant by the Mann–Kendall rank statistic test (significance inferred for 5%) is marked with hatched lines.**


We also investigated lightning trends simulated in different experiments with the global map (Fig. 7). Both the ECMWF-McCAUL and the CTH schemes show that lightning increased significantly in most parts of the Arctic region and decreased in some parts of the Southern Ocean during 1960–2014 (Figs. 7a, 7e). The significant lightning trends presented in Figs. 7a became nearly nonexistent when the climate simulations were fixed to 1959 (Figs. 7b, 7f), indicating the considerable effects of global warming on the trend of global lightning activities. Furthermore, the effects of past global warming and increases in AeroPEs on the lightning trends on the global map are displayed in Fig. 8. Figures 8a and 8c show that past global warming enhances lightning activities within the Arctic region and Japan, which is consistent with findings of an earlier study from which Japan thunder day data were reported (Fujibe, 2017). Figures 8a and 8c also show that historical global warming suppresses lightning activities around New Zealand and some parts of the Southern Ocean. Both lightning schemes demonstrated that the historical increases in AeroPEs suppress lightning activities in some parts of the Southern Ocean and South America. The ECMWF-McCAUL scheme also suggests that historical increases in AeroPEs suppress lightning activities by aerosol radiative effects in some parts of India and China, where AeroPEs increased dramatically during 1960–2014 because of rapid economic development and energy consumption. Many observation-based studies indicate that aerosols can invigorate lightning activities in some regions of China and India, typically under relatively clean conditions (e.g., AOD < 1.0), which is attributable to the aerosol microphysical effects (Wang et al., 2011; Zhao et al., 2017; Lal et al., 2018; Liu et al., 2020; Shi et al., 2020; Zhao et al., 2020). Therefore, a total positive effect of aerosol on historical lightning trends in China and India cannot be ruled out. We further provided the same figures as Figs. 7 and 8, but using different units

(fl. km$^{-2}$ yr$^{-2}$) in the supplementary information (Figs. S4 and S5). Figures S4 and S5 show that the absolute lightning trends
(fl. km$^{-2}$ yr$^{-2}$) and the effects of global warming and increases in AeroPEs on the absolute lightning trends are slight in high-
latitude regions but prominent in tropical areas.

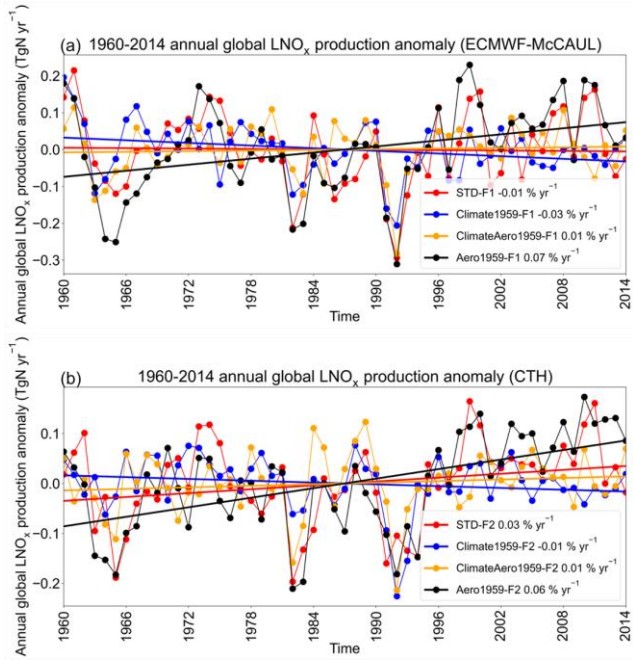


**Figure 9: Time-series data of 1960–2014 annual global LNO$_x$ production anomalies (TgN yr$^{-1}$) and their fitting curves simulated using the ECMWF-McCAUL scheme (a) and the CTH scheme (b). Trends of the fitting curves in percent per year are presented in the legends.**


**Table 4: A statistical summary of the trends shown in Fig. 9 by Mann–Kendall rank statistic and Sen's slope estimator. The time-series data of annual global LNO$_x$ production anomalies were estimated by Mann–Kendall rank statistic and Sen's slope estimator. The column "Trend" shows whether these are significant trends with the significance set as 5%, as well as the percentage trends in % yr$^{-1}$ estimated by linear regression. The "*p*-value" is calculated during Mann-Kendall trend test. "Slope" shows Sen's slope of trend. Q$_{min}$ and Q$_{max}$ respectively denote the lower and upper limits of the 95% confidence interval of Sen's slope.**


| Experiment | Trend | *p*-value | Slope | Q$_{min}$ | Q$_{max}$ |
|---|---|---|---|---|---|
| STD-F1 | No trend, -0.01 % yr$^{-1}$ | p > 0.05 | -0.0001 | -0.002 | 0.0018 |
| Climate1959-F1 | Decreasing, -0.03 % yr$^{-1}$ | p < 0.05 | -0.0011 | -0.0024 | -0.0001 |
| ClimateAero1959-F1 | No trend, 0.01 % yr$^{-1}$ | p > 0.05 | 0.0003 | -0.0008 | 0.0013 |
| Aero1959-F1 | Increasing, 0.07 % yr$^{-1}$ | p < 0.01 | 0.003 | 0.0011 | 0.0048 |
| STD-F1 – Climate1959-F1 | No trend, 0.02 % yr$^{-1}$ | p > 0.05 | 0.0009 | -0.0009 | 0.0025 |
| STD-F1 – Aero1959-F1 | Decreasing, -0.08 % yr$^{-1}$ | p < 0.01 | -0.003 | -0.004 | -0.0021 |

| | | | | | |
|---|---|---|---|---|---|
| STD-F2 | Increasing, 0.03 % $yr^{-1}$ | $p < 0.05$ | 0.0013 | 0.0001 | 0.0024 |
| Climate1959-F2 | No trend, -0.01 % $yr^{-1}$ | $p > 0.05$ | -0.0007 | -0.0014 | 0.0001 |
| ClimateAero1959-F2 | No trend, 0.01 % $yr^{-1}$ | $p > 0.05$ | 0.0005 | -0.0004 | 0.0015 |
| Aero1959-F2 | Increasing, 0.06 % $yr^{-1}$ | $p < 0.01$ | 0.0033 | 0.0019 | 0.0046 |
| STD-F2 – Climate1959-F2 | Increasing, 0.04 % $yr^{-1}$ | $p < 0.01$ | 0.0021 | 0.0006 | 0.0033 |
| STD-F2 – Aero1959-F2 | Decreasing, -0.03 % $yr^{-1}$ | $p < 0.01$ | -0.0019 | -0.0029 | -0.001 |


Trends in historical annual global $LNO_x$ emissions for different scenarios are generally consistent with trends in historical
global mean LFRs, as shown in Figs. 5a–5b and Fig. 9. This finding is not surprising because, as the lightning $NO_x$ emission
parameterizations introduced in Sect. 2.2 show, the simulated LFRs are linearly related to the simulated $LNO_x$ emissions in
our study. Comparison of the $LNO_x$ trends calculated from the STD and Climate1959 experiments showed that both
lightning schemes demonstrated that historical global warming (1960–2014) enhances the global $LNO_x$ trends toward
positive trends (0.02% $yr^{-1}$ – 0.04% $yr^{-1}$). Global warming effects on historical $LNO_x$ trends were evaluated as significant
using the Mann–Kendall rank statistic, with significance inferred for 5%, when using the CTH scheme, but not in the case of
the ECMWF-McCAUL scheme (see rows "STD-F1 – Climate1959-F1" and "STD-F2 – Climate1959-F2" in Table 4). As
shown in Table 4, the differences in global $LNO_x$ trends simulated by the STD and Aero1959 experiments indicate that the
increases in AeroPEs during 1960–2014 significantly suppress the global $LNO_x$ trends (-0.08% $yr^{-1}$ – -0.03% $yr^{-1}$). The
results presented in Fig. 9 and Table 4 imply that historical global warming and increases in AeroPEs can affect atmospheric
chemistry and can engender feedback by influencing $LNO_x$ emissions.
**3.3 Pinatubo volcanic eruption effects on historical lightning–$LNO_x$ trends**
We estimate the Pinatubo eruption effects on historical lightning–$LNO_x$ trends and variation by comparing the simulation
results of STD and Volca-off experiments. The simulated global mean LFRs by STD and Volca-off experiments are the
same until April 1991. They then begin to show differences from May 1991 (The time series of global mean LFRs is not
shown.). This result is reasonable because the Pinatubo volcanic perturbations are removed from SAC during June 1991 –
May 1996 in the Volca-off experiments by equation (11), and because the SAC of May 1991 used in CHASER is
interpolated between the SAC of April 1991 and June 1991.

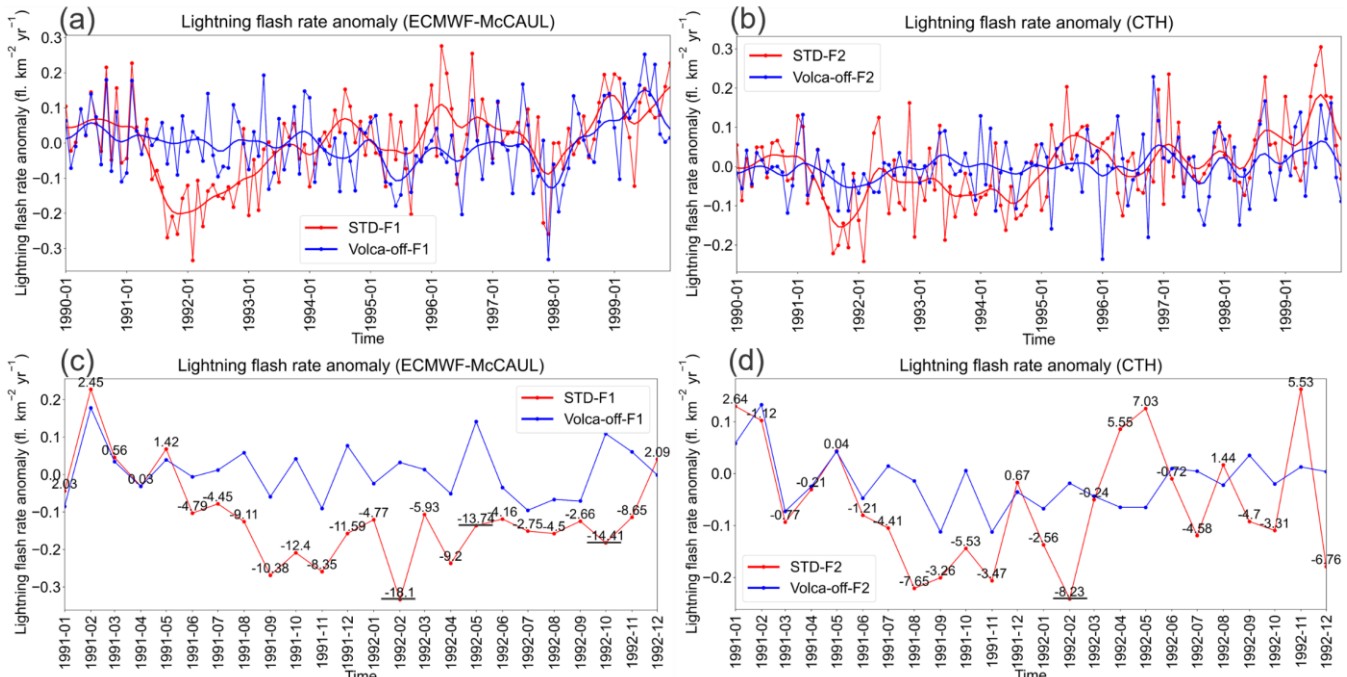

**Figure 10: Time series of LFR anomalies during 1990–1999 or during 1991–1992.** Panels (a) and (b) show the time series of LFR anomalies and their smoothed curves by 1-D Gaussian (Denoising) filter for 1990–1999. Panels (c) and (d) present the time series of LFR anomalies during 1991–1992. Values shown over the red lines in panels (c) and (d) are $Relative\_diff$ calculated using equation 12.

Figures 10c–10d portray the time series of LFR anomalies and $Relative\_diff$ (values over the red lines) during 1991–1992. $Relative\_diff$ are relative differences of the global mean LFR anomalies between STD and Volca-off experiments calculated using the following equation.

$$Relative\_diff = 100\% \times \frac{LFRA_{STD} - LFRA_{Volca-off}}{LFR_{Volca-off}} \qquad (12)$$

In the equation, $LFRA_{STD}$ represents global mean LFR anomalies simulated by STD-F1/F2 experiments. $LFRA_{Volca-off}$ denotes global mean LFR anomalies simulated by Volca-off-F1/F2 experiments. $LFR_{Volca-off}$ symbolizes global mean LFRs simulated by Volca-off-F1/F2 experiments.

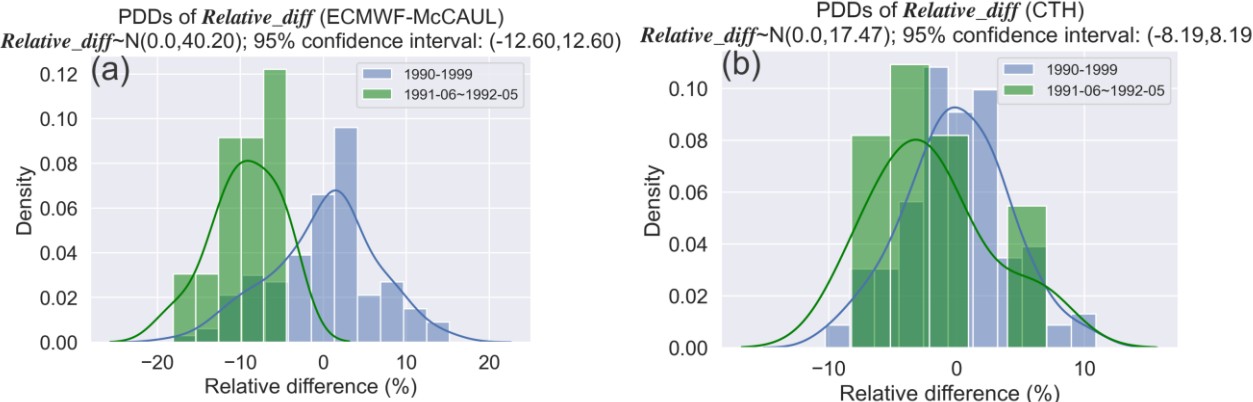

538

**Figure 11: Probability Density Distributions (PDDs) of *Relative_diff* obtained from monthly time-series data of *Relative_diff***

**during 1990–1999 and 1991-06 – 1992-05 (a year after the Pinatubo eruption). The 1990–1999 *Relative_diff* for both lighting**

**schemes are normally distributed with $N(\mu, \sigma^2)$ displayed in the titles of this figure. The 95% confidence interval of 1990–1999**

**_Relative_diff_ is also shown in the titles of this figure.**

543

The monthly time-series data of ***Relative_diff*** for 1990–1999 for both lightning schemes are calculated. The Probability

Density Distributions (PDDs) of ***Relative_diff*** spanning 1990–1999 and 1991-06 – 1992-05 are displayed in Fig. 11. The

1990–1999 ***Relative_diff*** presented in Fig. 11 (colored blue) are all normally distributed as determined by the

Kolmogorov–Smirnov test. The 95% confidence interval of 1990–1999 ***Relative_diff*** is calculated and shown in the titles

of Fig. 11. As displayed in Figs. 10c–10d, the underlined values (***Relative_diff***) exceeded the 95% confidence interval,

indicating significant differences in the calculated global mean LFR anomalies by STD and Volca-off experiments. In other

words, global lightning activities were suppressed significantly by the Pinatubo eruption during the first year after the

eruption. The PDDs of 1991-06 – 1992-05 ***Relative_diff*** (colored green in Fig. 11) shifted to the left compared to the

1990–1999 PDDs, indicating that global lightning activities were suppressed in the first year after the eruption.

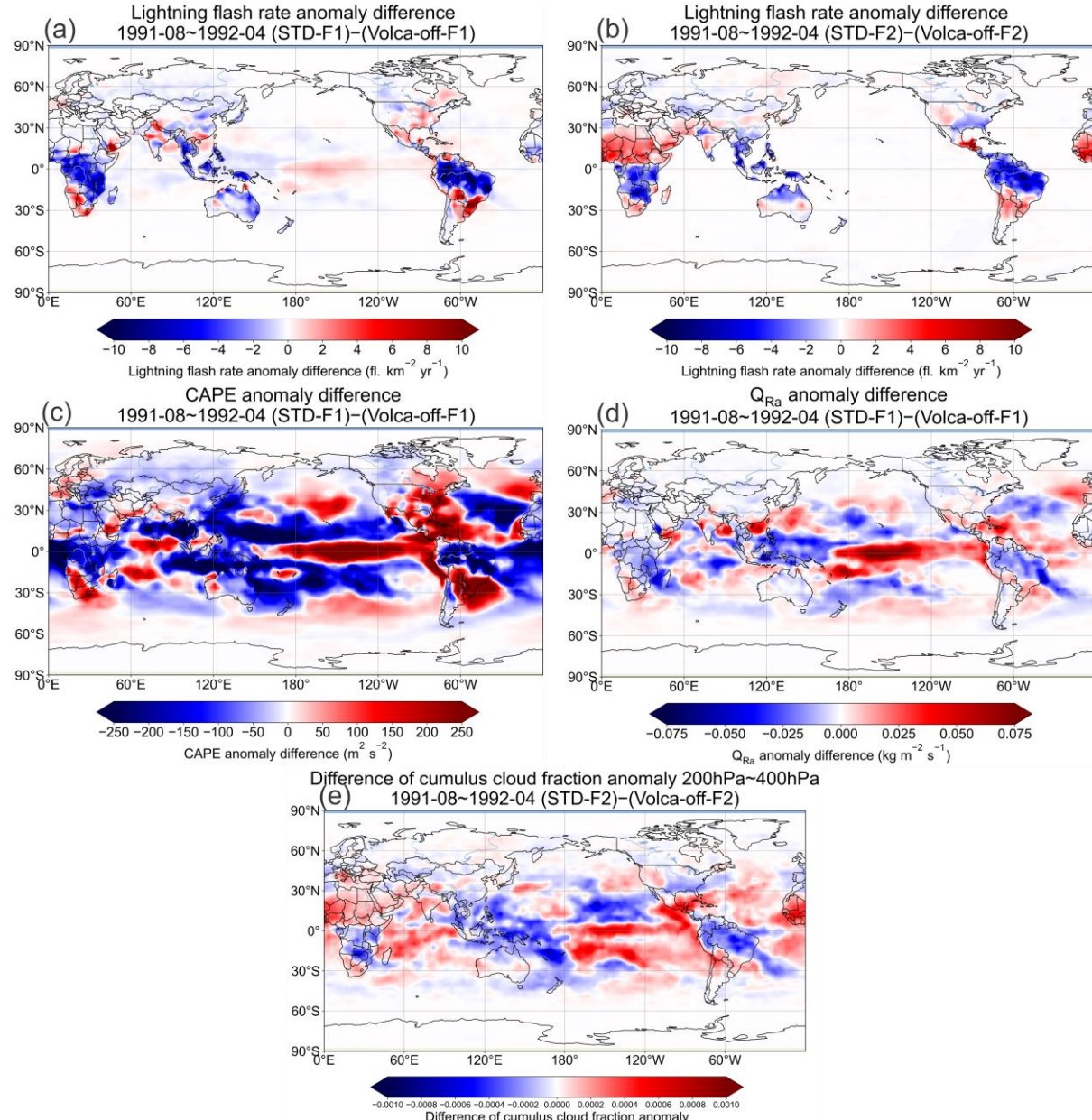


**Figure 12: 1991-08 – 1992-04 averaged LFR anomaly differences (a–b), CAPE anomaly differences (c), $Q_{Ra}$ anomaly differences**
**(d), and differences of 200 hPa – 400 hPa averaged cumulus cloud fraction anomaly between STD-F2 and Volca-off-F2**
**experiments (e) on the global map.**


Figures 12a–12b show 1991-08 – 1992-04 averaged LFR anomaly differences between STD and Volca-off experiments on
the global map. We found from Figs. 12a–12b that lightning activities are suppressed significantly within the three hotspots
of lightning activities (Central Africa, Maritime Continent, and South America) during 1991-08 – 1992-04, when the global

mean LFRs are found to be suppressed. To elucidate the potential reasons for the suppressed global lightning activities
during the first year after the Pinatubo eruption, we first investigated the 1991-08 – 1992-04 averaged differences in CAPE
and $Q_{Ra}$ anomaly between STD-F1 and Volca-off-F1 (Figs. 12c–12d) because lightning densities are computed with CAPE
and $Q_{Ra}$ by the ECMWF-McCAUL scheme. Results showed that the Pinatubo eruption can engender apparent reductions of
CAPE and $Q_{Ra}$ within tropical and subtropical terrestrial regions (typically three hotspots of lightning activities) where
lightning occurrence is frequent. These reductions constitute the main reason for the suppressed global lightning activities
during the first year after the Pinatubo eruption simulated by the ECMWF-McCAUL scheme. We also examined the 1991-
08 – 1992-04 averaged differences of 200 hPa – 400 hPa averaged cumulus cloud fraction anomaly between STD-F2 and
Volca-off-F2 on the global map (Fig. 12e). The cumulus cloud fractions of each model layer are used to weight the
calculated lightning densities from that layer by the CTH scheme, as explained in Sect. 2.2. As depicted in Fig. 12e and Fig.
S6, the Pinatubo eruption led to marked reductions in the middle to upper tropospheric cumulus cloud fractions during 1991-
08 – 1992-04 over three hotspots of lightning activities (Central Africa, Maritime Continent, and South America). As
displayed in Fig. 6h, the cumulus that reached the middle to upper troposphere is related closely to lightning formation.
Consequently, the simulated global lightning activities by the CTH scheme were also suppressed considerably during the
first year after the Pinatubo eruption.

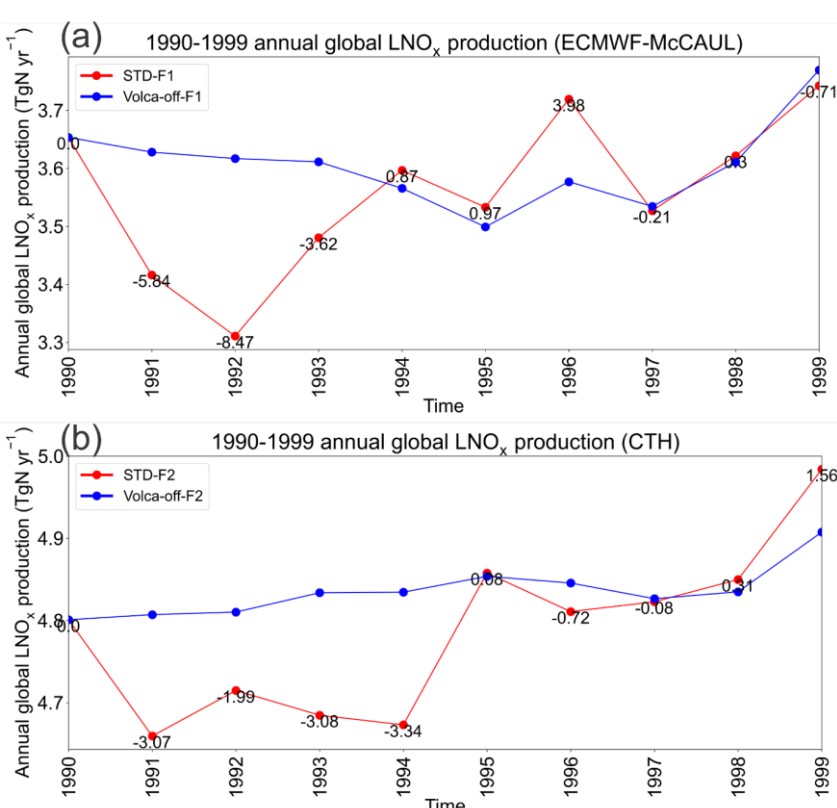


**Figure 13: 1990–1999 annual global LNO$_x$ emissions calculated from the STD and Volca-off experiments' outputs simulated using**
**the ECMWF-McCAUL scheme (a) and the CTH scheme (b). Values over the red lines represent the relative differences (%)**
**between the red lines and blue lines, calculated with respect to the blue lines.**

Aside from the global lightning activity suppression described earlier, the production of LNO$_x$ might also decrease after the
Pinatubo eruption. To explore this conjecture, we compared the LNO$_x$ emissions in STD and Volca-off experiments (Fig. 13).
In the case of the ECMWF-McCAUL scheme, the reduction of LNO$_x$ emissions caused by the Pinatubo eruption started in
1991 (5.84%) and continued until 1993, with the highest percentage reduction occurring in 1992 (8.47%) (Fig. 13a).
However, the CTH scheme showed a slightly different scenario of LNO$_x$ emissions reduction after the Pinatubo eruption.
The LNO$_x$ emissions are almost evenly reduced during 1991–1994 in the case of the CTH scheme (Fig. 13b). In conclusion,
our study indicates that the Pinatubo eruption can engender reductions in global LNO$_x$ emissions, which last 2–3 years.
However, there exists some uncertainty in evaluating the magnitude of the reductions: from 1.99% to 8.47% for the annual
percentage reduction found from our study.

The simulated reduced global LNO$_x$ emissions caused by the Pinatubo eruption might influence atmospheric chemistry
significantly. Most importantly, the reduced global LNO$_x$ emissions might reduce OH radical production and extend the
global mean tropospheric lifetime of methane against tropospheric OH radical, abbreviated hereinafter as the methane
lifetime. We investigated this point further by comparing the methane lifetime anomaly simulated by STD and STD-
rVolcaoff experiments. As introduced in Sect. 2.5, the settings of STD-rVolcaoff experiments are the same as those use for
STD experiments, except that they use the daily LNO$_x$ emission rates calculated from the Volca-off experiments. We
calculated the monthly CH$_4$ lifetime anomalies during 1990–1999 and $\Delta\tau_{CH_4}$ (the difference of CH$_4$ lifetime anomaly
between STD and STD-rVolcaoff experiments), which are shown in Figs. 14c–14d. Figures 14a–14b display the PDDs of
$\Delta\tau_{CH_4}$ monthly time series during 1990–1999. The $\Delta\tau_{CH_4}$ shown in Figs. 14a–14b are all normally distributed, as determined
using the Kolmogorov–Smirnov test. The 95% confidence interval of $\Delta\tau_{CH_4}$ is calculated and shown in the titles of Figs.
14a–14b. The annual global LNO$_x$ production averaged during 1990–1999 is 3.56 TgN yr$^{-1}$ for STD-F1 and 4.79 TgN yr$^{-1}$
for STD-F2. At this level of annual global LNO$_x$ production, we found that within the first two years after the Pinatubo
eruption, the $\Delta\tau_{CH_4}$ exceeded the 95% confidence interval simulated by both lighting schemes (1992-02 and 1992-04 in the
case of the ECMWF-McCAUL scheme; 1991-12 in the case of the CTH scheme). However, the widely cited range of annual
global LNO$_x$ production is 2–8 TgN yr$^{-1}$ (Schumann and Huntrieser, 2007). Presuming that $\Delta\tau_{CH_4}$ responds linearly to the
LNO$_x$ emission level, and that the annual global LNO$_x$ production is 8 TgN yr$^{-1}$, then the extension of the CH$_4$ lifetime
because of the reduced LNO$_x$ emissions can reach around 0.54 years for the ECMWF-McCAUL scheme. As a comparison,
ultraviolet shielding effects caused by stratospheric aerosols after the Pinatubo eruption led to the maximum increase of the
methane lifetime by about 0.6 years (Figs. 14c–14d).

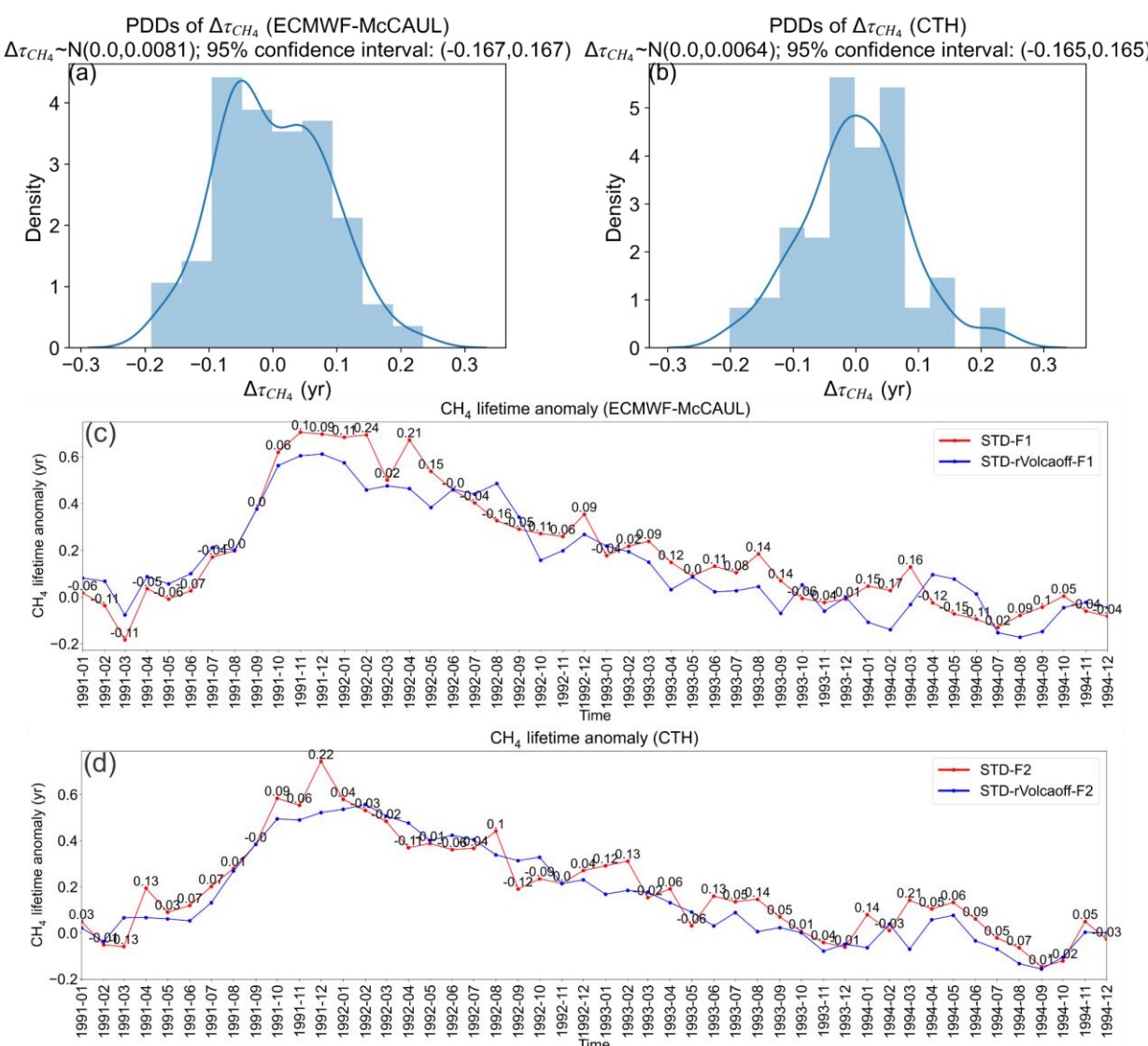


**Figure 14: Panels (a) and (b) show the Probability Density Distributions (PDDs) of $\Delta\tau_{CH_4}$ obtained from the monthly time series**


**data of $\Delta\tau_{CH_4}$ during 1990–1999. $\Delta\tau_{CH_4}$ represents the difference in CH₄ lifetime anomaly between STD and STD-rVolcaoff**


**experiments. The 95% confidence interval of $\Delta\tau_{CH_4}$ is also presented in the titles of panels (a)–(b). Panels (c) and (d) show monthly**


**time series of CH₄ lifetime anomalies simulated by STD-F1/F2 and STD-rVolcaoff-F1/F2 experiments. Values over the red lines**


**represent $\Delta\tau_{CH_4}$.**


**3.4 Model intercomparisons of LFR trends with CMIP6 model outputs**


The historical lightning trends demonstrated in our study are undoubtedly worth comparing with the results of other


chemistry–climate models or Earth system models. As introduced in Sect. 2.4, for comparison of the simulated LFR trends


and variations in our study with those of other CMIP6 models' outputs, we used all available LFR data from the CMIP6
CMIP Historical experiments from CESM2-WACCM (3 ensembles) (Danabasoglu, 2019), GISS-E2-1-G (9 ensembles)
(Kelley et al., 2020), and UKESM1-0-LL (18 ensembles) (Tang et al., 2019). Table S1 presents a complete list of the
ensemble members we used. It is noteworthy that the LFR data obtained from the three CMIP6 models described earlier are
calculated using the CTH scheme. The results of model intercomparisons of LFR trends and variations are displayed in Fig.

625 15.

As displayed in Figs. 15a–15b and Table 6, both the ECMWF-McCAUL and the CTH schemes (STD-F1/F2) simulated
almost flat global lightning trends (even the trend is estimated to be significant in the case of the CTH scheme (0.03 % $yr^{-1}$)),
but the ensemble mean obtained from another three CMIP6 models exhibit much larger significant increasing global
lightning trends (trends from 0.11% $yr^{-1}$ to 0.25% $yr^{-1}$). Many reasons underlie the differences in global lightning trends
simulated by CHASER in our study and by the three CMIP6 models, including the use of different methods to determine
SSTs/sea ice fields. Instead of using a coupled Atmosphere–Ocean general circulation model to calculate SSTs/sea ice fields
dynamically in the three CMIP6 models, CHASER uses the prescribed HadISST data (Rayner et al., 2003), which are based
on plenty of observational data. Changes in the global mean sea surface temperature anomaly during 1960–2014 (ΔSST)
obtained from STD-F1/F2 and CMIP6 model outputs are presented in Table 5. We also used the observation-based Extended
Reconstructed SST (ERSST) dataset (Huang et al., 2017) constructed by NOAA to evaluate the ΔSST obtained from
different models. The ΔSST calculated from ERSST during 1960–2014 is 0.549°C, which most closely approximates the
ΔSST obtained from STD-F1/F2. Considered from the perspective of SSTs/sea ice fields alone, the results (global lightning
trends) of our study are expected to be closer to the actual situation.

Actually, the three CMIP6 models simulated stronger global warming during 1960–2014 than CHASER in our study, as
displayed in Fig. S7 and Table 5. The CTH scheme is reported to respond positively to simulated global warming (Price and
Rind, 1994; Zeng et al., 2008; Hui and Hong, 2013; Banerjee et al., 2014; Krause et al., 2014; Clark et al., 2017). The
simulated stronger global warming by the three CMIP6 models is regarded as responsible for differences in simulated global
lightning trends between our study and the three CMIP6 models (Figs. 15a–15b and Table 6). We further investigated the
sensitivities of the global mean LFR anomaly change to the global mean surface temperature anomaly increase (% $°C^{-1}$)
obtained from CHASER and the three CMIP6 models. The sensitivities in percentage per degree Celsius are presented in
Table 5. Overall, even when using the same CTH scheme, the sensitivities (ΔLFR/ΔTS) simulated by the three CMIP6
models are higher than that simulated by CHASER in our study. This different sensitivity might be partially attributable to
the nonlinear relation between lightning response and climate change (Pinto, 2013; Krause et al., 2014). Compared to the
CTH scheme, the ECMWF-McCAUL scheme simulated a statistically non-significant negative sensitivity (ΔLFR/ΔTS),
which is attributable to the stronger suppression of positive global lightning trends caused by increases in AeroPEs simulated
using the ECMWF-McCAUL scheme.

**Table 5: Changes in global mean surface temperature anomaly (ΔTS), global mean sea surface temperature anomaly (ΔSST),**
**global mean lightning flash rate anomaly (ΔLFR), and the rate of change of LFR anomaly corresponding to each degree-Celsius**
**increase in global mean surface temperature anomaly (ΔLFR/ΔTS) obtained from STD-F1/F2 and CMIP6 model outputs. The**
**change of ΔSST obtained from the ERSST dataset is also shown in this Table. Changes were obtained by calculating the difference**
**between the rightmost and leftmost points of the approximating curve for the 1960–2014 time-series data.**

| Model/experiment/dataset | ΔTS (°C) | ΔSST (°C) | ΔLFR (%) | ΔLFR/ΔTS (% °C$^{-1}$) |
|---|---|---|---|---|
| STD-F1 | 0.593 | 0.428 | -0.272 | -0.46 |
| STD-F2 | 0.563 | 0.432 | 1.497 | 2.66 |
| CESM2-WACCM | 1.245 | 1.077 | 13.758 | 11.05 |
| GISS-E2-1-G | 0.810 | 0.677 | 7.248 | 8.95 |
| UKESM1-0-LL | 1.141 | 0.999 | 5.942 | 5.21 |
| ERSST | — | 0.549 | — | — |


Figures 15d–15e affirm that the global lightning variation simulated by our study is basically within the full ensemble range
of GISS-E2-1-G and UKESM1-0-LL. After the Pinatubo eruption, as described in Sect. 3.3 of this report, the GISS-E2-1-G
and UKESM1-0-LL models also manifest significant suppression of global lightning activities, but the CESM2-WACCM
model shows no such phenomenon. The commonalities and differences in global lightning trends and variations found in the
model intercomparisons imply that great uncertainties existed in past (1960–2014) global lightning trend simulations. Such
uncertainties deserve to be investigated further.

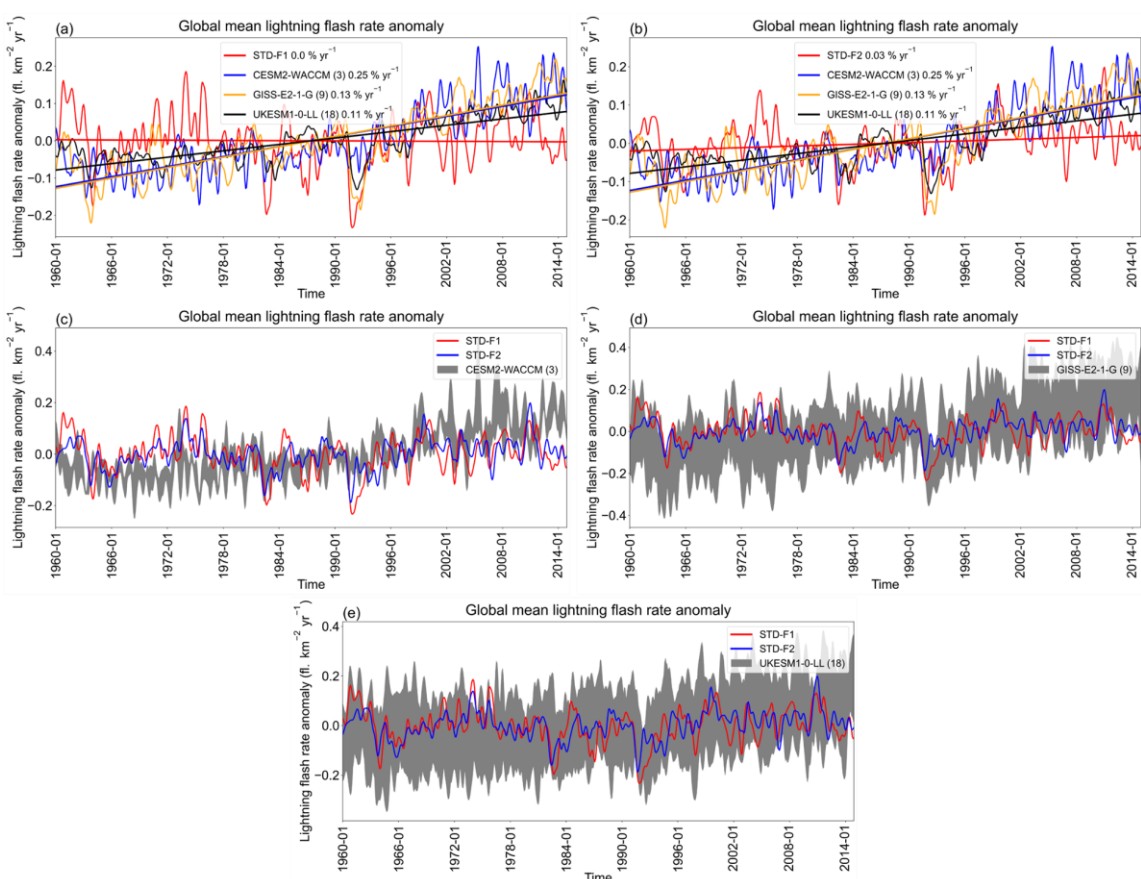


**Figure 15: Comparisons of simulated global mean LFR anomalies found in our study (CHASER) and found using other CMIP6**
**models. All the figures are created based on the monthly time-series data of global mean LFR anomalies with a 1-D Gaussian**
**(Denoising) filter applied. For CMIP6 models, the ensemble mean is shown as the solid line, and the full ensemble range is shown**
**as grey shading (c–e). Fitting curves and the trends of fitting curves (% yr$^{-1}$) are also given in (a–b).**

**Table 6: A statistical summary of the trends shown in Figs. 15a–15b by Mann–Kendall rank statistic and Sen's slope estimator.**
**The time-series data of global mean LFR anomalies were estimated by Mann–Kendall rank statistic and Sen's slope estimator. The**
**column "Trend" shows whether these are significant trends with the significance set as 5%, as well as the percentage trends in %**
**yr$^{-1}$ estimated by linear regression. The "$p$-value" is calculated during Mann-Kendall trend test. "Slope" shows Sen's slope of**
**trend. $Q_{min}$ and $Q_{max}$ respectively denote the lower and upper limits of the 95% confidence interval of Sen's slope.**

| Experiment/model | Trend | $p$-value | Slope | $Q_{min}$ | $Q_{max}$ |
|---|---|---|---|---|---|
| STD-F1 | No trend, 0.0 % yr$^{-1}$ | $p > 0.05$ | 0.0 | -0.0001 | 0.0 |
| STD-F2 | Increasing, 0.03 % yr$^{-1}$ | $p < 0.01$ | 0.0001 | 0.0 | 0.0001 |
| CESM2-WACCM | Increasing, 0.25 % yr$^{-1}$ | $p < 0.01$ | 0.0004 | 0.0003 | 0.0004 |

| GISS-E2-1-G | Increasing, 0.13 % yr$^{-1}$ | $p < 0.01$ | 0.0004 | 0.0004 | 0.0004 |
| UKESM1-0-LL | Increasing, 0.11 % yr$^{-1}$ | $p < 0.01$ | 0.0002 | 0.0002 | 0.0003 |

## 4 Discussion and Conclusions

We used two lightning schemes (the CTH and ECMWF-McCAUL schemes) to study historical (1960–2014) lightning–LNO$_x$ trends and variations and their influencing factors (global warming, increases in AeroPEs, and Pinatubo eruption) within the CHASER (MIROC) chemistry–climate model. The CTH scheme, which is the most widely used lightning scheme, nevertheless lacks a direct physical link with the charging mechanism. The ECMWF-McCAUL scheme is a newly developed process-based/ice-based lightning scheme with a direct physical link to the charging mechanism.

With only the aerosol radiative effects considered in the lightning–aerosols interaction, both lightning schemes simulated almost flat trends of global mean LFR during 1960–2014 (no trend is detected in the case of the ECMWF-McCAUL scheme, but a slightly significant increasing trend is detected in the case of the CTH scheme). Reportedly, because the aerosol microphysical effects can enhance lightning activities (Yuan et al., 2011; Wang et al., 2018; Liu et al., 2020), our study might underestimate the increasing trend of global mean LFR (our study only considered the aerosol radiative effects in aerosol–lightning interactions). Further research is anticipated, with consideration of the effects of aerosol microphysical effects on long-term lightning trends. Moreover, both lightning schemes manifest that past global warming enhances the historical trend of global mean lightning density toward the positive direction (around 0.03% yr$^{-1}$ or 3% K$^{-1}$). However, past increases in AeroPEs exert the opposite effect to the lightning trend (-0.07% yr$^{-1}$ – -0.04% yr$^{-1}$). The effects of the increased AeroPEs on the lightning trend only over land regions expand to -0.10% yr$^{-1}$ – -0.05% yr$^{-1}$, which implies that the effects are more significant over land regions. We obtained similar results for the historical global LNO$_x$ emissions trend, which indicates that historical global warming and increases in AeroPEs can affect atmospheric chemistry and engender feedback by influencing LNO$_x$ emissions. Although the CTH and ECMWF-McCAUL schemes use different parameters to simulate lightning, both lightning schemes indicate that the enhanced global convective activity under global warming is the main reason for the increase in lightning–LNO$_x$ emissions. In contrast, the increases in AeroPEs have decreased lightning–LNO$_x$ emissions by weakening the convective activity in the lightning hotspots. By analyzing the simulation results on the global map, we also found that the effects of historical global warming and increases in AeroPEs on lightning trends are heterogeneous across different regions. Our results indicate that historical global warming enhances lightning activities within the Arctic region and Japan but suppresses lightning activities around New Zealand and some parts of the Southern Ocean. Both lightning schemes demonstrated that the historical increases in AeroPEs suppress lightning activities in some parts of the Southern Ocean and South America. The ECMWF-McCAUL scheme also suggests that historical increases in AeroPEs suppress lightning activities in some parts of India and China when only the aerosol radiative effects are considered.

This finding is plausible because both countries experienced dramatic increases in AeroPEs during 1960–2014 because of
rapid economic growth.

Furthermore, this report is the first describing significant suppression of global lightning activity during the first year after
the Pinatubo eruption, which is indicated in both lightning schemes (global lightning activities decreased by up to 18.10%
simulated by the ECMWF-McCAUL scheme). This finding is mainly attributable to the Pinatubo eruption weakening of the
convective activities within the hotspots of lightning, which in turn decreased $Q_{Ra}$ and middle-level to high-level cumulus
cloud fractions in these regions. The simulation results also indicate that the Pinatubo eruption can engender reductions in
global $LNO_x$ emissions, which last 2–3 years. However, some uncertainty exists in evaluating magnitude of these reductions
(from 1.99% to 8.47% for the annual percentage reduction in our study). The case study of the Pinatubo eruption in our
research indicates that other large-scale volcanic eruptions can also engender significant reduction of global lightning
activities and global-scale $LNO_x$ emissions.

Lastly, we compared the global lightning trends demonstrated in our study with the outputs of three CMIP6 models:
CESM2-WACCM, GISS-E2-1-G, and UKESM1-0-LL. We used all available LFR data from the CMIP6 CMIP historical
experiments from the three models described above. The three CMIP6 models suggest significant increasing trends in
historical global lightning activities, which differs from the findings of our study in the magnitude of lightning trends. Unlike
the three CMIP6 models which use a coupled Atmosphere–Ocean general circulation model to calculate SSTs/sea ice fields
dynamically, our study (CHASER) uses the prescribed HadISST SSTs/sea ice data, which more closely reflect the actual
situation. Therefore, we believe that the results (the historical global lightning trends) obtained from our study (CHASER)
more closely approximate the actual situation. However, model intercomparisons of global lightning trends still indicate that
considerable uncertainties exist in historical (1960–2014) global lightning trend simulations, and that such uncertainties
deserve further investigation.
**Code availability**
The source code for CHASER to reproduce results obtained from this work is obtainable from the repository at
https://doi.org/10.5281/zenodo.5835796 (He et al., 2022a).
**Data availability**
The LIS/OTD data used for this study are available from https://ghrc.nsstc.nasa.gov/hydro/?q=LRTS (last access: 11 January
2022). The CMIP6 model outputs (LFR and surface temperature) used for this study are available from
https://aims2.llnl.gov/search (last access: 1 February 2023). The Extended Reconstructed SST data used for this study are
available from https://www.ncei.noaa.gov/products/extended-reconstructed-sst (last access: 27 March 27 2023).

**Author contributions**

YFH conducted all simulations, interpreted the results, and wrote the manuscript. KS developed the CHASER (MIROC)
model code, conceived the presented idea, and supervised the findings of this work and the manuscript preparation.

**Competing interests**

The authors declare that they have no conflict of interest.

**Acknowledgments**

This research was supported by the Global Environment Research Fund (S–12 and S–20) of the Ministry of the Environment
(MOE), Japan, and JSPS KAKENHI Grant Numbers: JP20H04320, JP19H05669, and JP19H04235. This work was
supported by the Japan Science and Technology Agency (JST) Support for Pioneering Research Initiated by the Next
Generation (SPRAING), Grant Number JPMJSP2125. The author would like to take this opportunity to thank the
"Interdisciplinary Frontier Next-Generation Researcher Program of the Tokai Higher Education and Research System." The
simulations were completed using the supercomputer (NEC SX-Aurora TSUBASA) at NIES (Japan). We thank NASA
scientists and staff for providing LIS/OTD lightning observation data. We acknowledge the World Climate Research
Programme, which coordinated and promoted CMIP6 through its Working Group on Coupled Modelling. We extend our
sincere gratitude to the climate modelling groups for producing and providing their model outputs, to the Earth System Grid
Federation (ESGF) for archiving the data and providing free downloads, and to the multiple funding agencies that have
supported the CMIP6 as well as the Earth System Grid Federation. We also thank Ms. Do Thi Nhu Ngoc for her assistance in
downloading the CMIP6 model outputs.

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
