# Peer review of "Historical (1960–2014) lightning and LNOx trends and their controlling factors in a chemistry–climate model"

_EGUsphere, 2023_

## Author Response (AR1)

Dear referees, we sincerely appreciate the efforts you have made to review our manuscript and to give constructive comments related to it. We have carefully revised our manuscript based on your comments. Hereinafter, I shall respond to the Referees' comments point-by-point and explicitly point out changes with line numbers in the revised manuscript. The line number (or Section or Figure number, etc.) in blue is the line number (or Section or Figure number, etc.) in the preprint and the line number (or Section or Figure number etc.) in orange is the line number (or Section or Figure number, etc.) in our revised manuscript. Additionally, we use LFR as an abbreviation of the "lightning flash rate".

**Referee #1 comment:**

This manuscript presents the results of model simulations of lightning flash rate and resulting lightning $NO_x$ emissions over the period during 1960 to 2014 using the CHASER chemistry and climate model. The authors attempt to isolate the effects of global warming, changes in aerosol abundance, the Pinatubo volcanic eruption on flash rates. They are very successful in doing that with regard to warming and Pinatubo. However, the CHASER model only includes the radiative effect of aerosols, and the microphysical effects of aerosols in convective clouds are not included. On one hand, I could see that this could be a reason to reject the manuscript. However, on the other hand, it is interesting to see the effect on flash rates of the aerosol changes to atmospheric radiation. If the paper is allowed to proceed to publication, the authors must put more emphasis throughout the paper on the fact that the microphysical effects of aerosol on flash rates are neglected in this analysis. Yes, a caution about the aerosol results is written into Section 4 (Discussion and Conclusions), but it also needs to be prominent in the abstract and in the Results section. Inclusion of the microphysical effect could cause changes in flash rate of opposite sign compared to the results reported in this paper.

**Author comment:** We acknowledge that the absence of considering aerosol microphysical effects in convective clouds might be one deficiency of our study. However, great uncertainty still exists related to microphysical processes in convective clouds (Tao et al., 2012; Seinfeld et al., 2016; Heikenfeld et al., 2019). Introducing it does not necessarily improve the global lightning distribution and trend. Moreover, the current model ensembles in the latest IPCC/CMIP6 simulations mostly consider CCN only for large-scale (grid-scale) clouds, especially for chemistry–climate models, just as our present study. As also described in your comment, we think that reporting our results to show possible changes in lightning and $LNO_x$ using one of the state-of-the-art global chemistry–climate models is valuable. As you have suggested, we have also emphasized that the microphysical effects of aerosol on LFR are not considered in our study (L22-23, L384-385, L412-414, L428, L454-460, L628-632, L648).

**Referee #1 comment:**

line 54: add Romps et al. (2014) reference; change "minor" to "other"

**Author comment:** We have revised the text as you have suggested (L55).

**Referee #1 comment:**

lines 90-92: over what years?

**Author comment:** The comparison was conducted during 2007–2011; we have added this information to L95.

**Referee #1 comment:**

lines 151 - 152: I understand that this formulation takes into account convective clouds of various heights within a model grid cell. However, $H_i <$ freezing level should not produce any lightning. Is this taken into account?

**Author comment:** First, many studies have indicated that warm clouds can also produce lightning (Rossby, 1966; Takahashi, 1975; Vaughan and Boeck, 1998). Secondly, even if lightning flashes originate from a place higher than the 0°C

isotherm, the lightning channels can still reach a place much lower than 0°C isotherm. Moreover, upward lightning flashes can originate from the surface. Those kinds of lightning flashes described above can be allocated to the model layers under 0°C isotherm. Additionally, we have not found from the paper describing the CTH scheme (Price and Rind, 1992) that the LFR should be set to 0 if the cloud top is under 0℃ isotherm.

**Referee #1 comment:**

line 166: If $Q_{Ra}$ includes cloud ice, it should not be described as "precipitating ice"

**Author comment:** Thank you very much for your suggestion. We have avoided using the term "precipitating ice"; instead, we used "total volumetric amount of the hydrometeors of three kinds (graupel, snow, and cloud ice)" (please see L142, L163-164, L171-172).

**Referee #1 comment:**

line 191: emissions. For example, Allen et al. (2019) and Bucsela et al. (2019) found that lower LNO$_x$ production per flash was associated with larger flash rates.

**Author comment:** Thank you for providing this information.

**Referee #1 comment:**

lines 196 - 200: these should all be written in the past tense since OTD and TRMM/LIS are no longer operational

**Author comment:** We have revised the verb tense as you have suggested (L201–208).

**Referee #1 comment:**

line 203: is LRTS the mean for each day over the 20-year period?

**Author comment:** Yes, the low resolution time series (LRTS) gives a multi-year time series of the smoothed daily flash rate. An appropriate number of days at the beginning and end were discarded to allow for smoothing (Cecil et al., 2014).

**Referee #1 comment:**

line 235: This equation needs more explanation. I'm not familiar with this formulation.

**Author comment:** We now have added more explanations on this equation (please see L234-236 and L247-254). Please also note that **we have optimized the methodology to remove the Pinatubo perturbation** in Stratospheric Aerosol Climatology (SAC). The new methodology can remove the Pinatubo perturbation in SAC more sufficiently and can also maintain the data consistency (please see below, the period 2001–2010 is far from the time of Pinatubo eruption, but 1986-06 – 1991-05 and 1996-06 – 2001-05 are closer to the time of Pinatubo eruption). We now use the following equations to process SAC **from June 1991 to May 1996**.

$$SAC_{no\_pinatubo} = \begin{cases} SAC_{background}, |SAC_{raw} - SAC_{background}| > 1.96\sigma, \\ SAC_{raw}, |SAC_{raw} - SAC_{background}| \leq 1.96\sigma \end{cases}$$

In that equation, $SAC_{no\_pinatubo}$ denotes the stratospheric aerosol climatological data as input data for Volca-off-F1/F2 experiments, $SAC_{background}$ represents the stratospheric background aerosol climatological data (For this study, $SAC_{background}$ represents the corresponding temporal averaged values of the NASA GISS and CCMI stratospheric aerosol dataset **during 1986-06 – 1991-05 and 1996-06 – 2001-05**, when the time is close to the eruption and the stratosphere was less affected by volcanic eruptions). Also, $SAC_{raw}$ stands for the original values of NASA GISS and CCMI stratospheric aerosol dataset during June 1991 – May 1996. Moreover, $\sigma$ symbolizes the standard deviations of stratospheric background aerosol climate data (For this study, $\sigma$ denotes the corresponding standard deviations of NASA GISS and

CCMI stratospheric aerosol dataset **during 1986-06 – 1991-05 and 1996-06 – 2001-05**). Instead of using the three–sigma rule, we found that using the 95% confidence interval ($1.96\sigma$) is more appropriate when $SAC_{background}$ and $\sigma$ are calculated based on the data during 1986-06 – 1991-05 and 1996-06 – 2001-05.

**Referee #1 comment:**

Figure 1: Both schemes put too many flashes over oceans. The colors cover the factor of 5 range from 10 to >50 are too similar. It is difficult to differentiate them. Need to use a different color scheme.

**Author comment:** We now have changed the color scheme in the revised manuscript (please see Fig. 1).

**Referee #1 comment:**

Figure 2: Earlier in the paper it says that LIS data only extends to ±38 degrees latitude. Why is this figure using ±41.25 degrees?

**Author comment:** From a paper describing the LIS/OTD dataset, the LIS collected data for ±38° latitude (Cecil et al., 2014). However, somehow there exist valid values (observations) within 38°N~41.25°N and 38°S~41.25°S in the LIS/OTD 2.5 Degree Low Resolution Time Series (LRTS) dataset. For consistency, we have now changed the range of spatial mean in Fig. 2 to ±37.5° latitude.

**Referee #1 comment:**

Figure 5f: Why are contributions to lightning coming from the shallow clouds (below the 600 hPa level)? Ice is needed for lightning.

**Author comment:** As described above, many results of studies suggest that warm clouds can produce lightning (Rossby, 1966; Takahashi, 1975; Vaughan and Boeck, 1998). Moreover, even if lightning flashes originate from a cold cloud, the lightning channels can still reach much lower places. Depending on the lightning flash type, upward lightning flashes can even originate from the surface. Even though we do not distinguish different types of lightning, such kinds of lightning flashes described above can be allocated to the model layers under the 0°C isotherm. Additionally, we have not found from the paper describing the CTH scheme (Price and Rind, 1992) that the LFR should be set to 0 if the cloud top is under 0℃ isotherm .

**Referee #1 comment:**

line 369-370: Why would that necessarily reduce $Q_{Ra}$? Clouds could be taller and have similar $Q_{Ra}$ as before global warming.

**Author comment:** Thank you very much for your suggestion. Yes, actually "the shifting of 0 – -25°C region to the higher altitude" cannot always explain the reduced $Q_{Ra}$ adequately. Based on an earlier study (Romps, 2019) (please see Fig. 4 in their study), global warming exerts negligible effects on the vertical profile of total solid (cloud ice, snow, and graupel) mass flux ($\varphi$). In other words, $\varphi$ is a roughly invariant function of the isotherm under global warming. This phenomenon is also shown by our study. However, global warming can engender the increase of CAPE as well as updraft velocity ($w$) within the charge separation regions. Also, $\varphi$ is calculated as a function of the total solid mass mixing ratio ($q$) multiplied by $w$. If $\varphi$ is almost constant but if $w$ increases, then one can expect a decrease of $q$. This decrease can explain why $Q_{Ra}$ decreases under global warming. Because the mechanism of why $Q_{Ra}$ decreases under global warming is not the target of this study, we prefer to cite the relevant literature to explain it (L402-404).

**Referee #1 comment:**

lines 374 - 378: Here is one of the locations where the lack of consideration of the microphysical effect of aerosols on hydrometeors and flash rates needs to be mentioned. This effect could increase hydrometeors, $Q_{Ra}$, and flash rate.

**Author comment:** Thank you very much for your suggestion. As you have suggested, we have highlighted the possible effects of the absence of consideration of aerosol microphysical effects on $Q_{Ra}$ and lightning trends (L412–L414).

**Referee #1 comment:**

line 386: need to compare 1960 and 2014 in Figure 5f

**Author comment:** Thank you for your suggestion. We have compared the vertical profiles of the relative contributions in 1960 and 2014 in Fig. 6f.

**Referee #1 comment:**

line 390: again, indicate that the decrease in cumulus reaching 200 hPa with increases in AeroPEs is only considering radiative effects.

**Author comment:** We have revised the related text to emphasize that we have only considered aerosol radiative effects (L428).

**Referee #1 comment:**

line 417: Here again, the aerosol results on flashes from the model may be opposite of what has been observed. There have been numerous papers looking at observations over China that show increasing flashes as aerosol increased and decreasing flashes as aerosol has declined.

**Author comment:**

Yes, as you have suggested, we have added relevant information to discuss differences between observation-based past studies and our results (L456–L459).

**Referee #1 comment:**

Figure 10: Is there greater density of negative relative differences? If so, that should be mentioned.

**Author comment:** We appreciate your good suggestion. However, we found that the percentages of negative relative differences are slightly less than 50% for both lightning schemes.

**Response to Anonymous Referee #2 comment,**

**Referee #2 comment:**

I have read this paper on historical modelled trends in lightning activity from 1960-2014. Unfortunately, I have many problems with the paper that are described below. The paper looks at three factors that could have impacted lightning activity (and LNO$_x$) over this period. The first is the increasing temperatures, the second is the increasing aerosol loading of the atmosphere, and the third is the Pinatubo volcanic eruption in 1991. The authors use 2 different parameterizations of lightning in their model.

**Author comment:** We sincerely thank the reviewer for the dedication of your time reviewing our manuscript and for pointing out its shortcomings. We specifically clarified each point. To improve the manuscript, we have revised our manuscript accordingly.

**Referee #2 comment:**

The model results show no real change in the lightning activity for changes in temperature and aerosols (different to other

simulations published by others and CMIP6), irrelevant of the parameterization used.

**Author comment:** Referring to a paper entitled "Parameterization-based uncertainty in future lightning flash density" (Clark et al., 2017), the Community Atmosphere Model ver. 5 (CAM5) with prescribed SSTs and sea ice field with eight different lightning schemes (including four CTH-based schemes and schemes based on convective precipitation rate and convective mass flux) also simulated almost flat historical (1960–2014) lightning trends (please see Fig. 3a in this paper). We think the difference between our results and other CMIP6 models is mainly attributable to the significant biases in SSTs and sea ice fields simulated by those CMIP6 models. Moreover, no observations of global lightning changes over 1960–2014 that could be used to validate the model simulation. Apparently, no one knows the real lightning changes. We believe that our study can be regarded as an important addition to such a perspective on past lightning activity at the global scale.

**Referee #2 comment:**

The trends in lightning over this period are not significant, and given the small increases in temperature we may not expect any significant changes. But when looking at the model lightning response spatially (Figure 6) the anomalies are in the polar regions mainly, and not where we normally see lightning. Other studies predict increases in lightning in the tropics.

**Author comment:** It is worth noting that Figs. 6–7 show the percentage lightning trend (% yr$^{-1}$) and that Figs. S3–S4 show the absolute lightning trend (fl. km$^{-2}$ yr$^{-2}$). Because lightning seldom strikes over polar regions, a small amount of absolute increase or decrease in LFR can lead to large values of percentage changes (% yr$^{-1}$). A study based on World Wide Lightning Network (WWLLN) data also indicate the Arctic as an extremely sensitive region of lightning to global warming (Holzworth et al., 2021). Figure S3 illustrates that LFR also increased or decreased considerably within tropical regions if one views the absolute lightning trend (fl. km$^{-2}$ yr$^{-2}$). We have explicitly pointed out the above-mentioned information (L460-463).

**Referee #2 comment:**

More worrying is that the increase in lightning due to temperature increases (greenhouse gases) and aerosol increases show the same spatial response, while the aerosol forcing is very different in location to the greenhouse gas forcing. Both simulations show strong positive anomalies in the Arctic and Antarctica, and strong negative anomalies around Antarctica, regardless of whether the forcing is from aerosol changes or temperature changes (greenhouse changes). This raises a red flag since the distribution of the changes should be spatially different. And in both scenarios little change occurs in the tropics. NOT realistic.

**Author comment:**

We do not think this result derives from using a flawed experiment setup or technical errors. As demonstrated by a study described in a paper entitled "Similar spatial patterns of global climate response to aerosols from different regions" (Kasoar et al., 2018), the climate model shows similar spatial patterns of the temperature response, which are irrelevant to the location of the (aerosol) forcing (please see Fig. 3 in the paper described above). Moreover, the spatial patterns of the temperature response are similar whether the forcing comes from aerosols or GHGs (please see Figs. S1 and S3 in the paper described above). Earlier studies suggest that regionally similar spatial patterns of lightning response as displayed in our study would normally be expected in climate model simulations.

Regarding the issue that little change occurs in the tropics, the absolute changes (fl. km$^{-2}$ yr$^{-2}$) in LFR are actually significant within tropical regions, as displayed in Figs. S4–S5 (Figs. S3–S4). The changes in LFR look not apparent in Fig. 7 (Fig. 6) simply because we used percentage differences (% yr$^{-1}$) in Fig. 7 (Fig. 6).

**Referee #2 comment:**

How are the global trends calculated? From the global absolute lightning changes? Or the changes in each pixel (%) averaged over the globe. The results imply that the lightning in this model is NOT sensitive to changes in temperature and aerosols.

**Author comment:** The global trends in Fig. 4 (Fig. 5) are calculated from the global absolute lightning changes.

**Referee #2 comment:**

However, the aerosol effect is only the radiative effect. The authors state they do not include the microphysical effect of increasing CCN since they simply cannot in this model. So simulating the aerosol impact with no microphysics is misleading, and not useful for other researchers. It is clear that having more aerosols will cool the surface and hence should reduce convection. But this is not a new result. Others have already shown this. So why publish these results if not realistic without including all the process linking aerosols and lightning?

**Author comment:** Yes. Including the microphysical effects of aerosols on convective clouds might be necessary for better simulation. However, **there are still large uncertainties in the microphysical processes in convective clouds** (Tao et al., 2012; Seinfeld et al., 2016; Heikenfeld et al., 2019**). Therefore, introducing it does not necessarily improve the global lightning distribution and trend**. Moreover, the current model ensembles in the latest IPCC/CMIP6 simulations mostly consider CCN only for large-scale (grid-scale) clouds, especially for chemistry–climate models, just as our present study. Only a few models include some kinds of microphysical processes of aerosols for convective (sub-grid-scale) clouds. As also suggested by Referee #1, we consider it is still important to report our experiment results to show possible changes in lightning and $LNO_x$ using a state-of-the-art global chemistry-climate model.

**Referee #2 comment:**

Furthermore, unlike what is expected, the aerosol simulations show an INCREASE in lightning over time. How do you explain this?

**Author comment:** I think the "aerosol simulations" you described are the Aero1959 experiments. Aero1959 is the experiment with AeroPEs fixed to 1959 throughout the simulation period, not the case with annually varying AeroPEs. We used the annually varying AeroPEs data (CMIP6 forcing datasets) in the STD experiments, which means AeroPEs increased dramatically in the STD experiments. During the simulation period, the global aerosol radiative effects increased in STD experiments but remained nearly constant in Aero1959 experiments. Because we simulated almost flat global lightning trends in STD experiments, we can see increasing global lightning trends simulated by Aero1959 experiments.

**Referee #2 comment:**

More worrying is the temporal change in the simulated lightning from the two parameterizations, especially F1 (Figure 9 e and f). The F1 parameterization shows two maxima in global lightning every year, while the F2 parameterization shows only one. From my knowledge the LIS/OTD data show one maximum per year in the northern hemisphere summer. If so, how can you use the F1 parameterization in this study when it cannot reproduce the seasonal variability of global lightning? Without explaining the observed temporal variability of global lightning we cannot trust the results of this parameterization. And if the F1 parameterization agrees with LIS/OTD, then F2 does not.

**Author comment:** Thank you very much for your comment. We also think the reproducibility of LFR seasonal variation in CHASER requires some special focus. Therefore, we have added an explanation of the validation of simulated LFR seasonal variation by LIS/OTD observations (please see L319–342, Fig. 3, Fig. S1). From Fig. 3, both lightning schemes

captured the peak during JJA, but we also noticed the overestimation of LFR during DJF. From comparison of the LFR global distribution in different seasons during 1996–2003 from LIS/OTD observations and STD experiment outputs (please see Fig. S1), we found that CHASER well captured the spatial distribution of LFR in all four seasons when compared against LIS/OTD observations. As displayed in Fig. S1, both lightning schemes can clearly capture the shifting of lightning distributions along the latitude band in different seasons. The spatial correlation coefficients ($R$) between observations and simulations are highest ($R$=0.80 for both lightning schemes) in DJF (please see Fig. S1), indicating CHASER's extremely high capability to reproduce the LFR spatial distribution in DJF. As displayed in the first row of Fig. S1, overestimation of LFR by F1/F2 during DJF occurs primarily because of the overestimation of LFR within the Maritime Continent and South America, but this result might also be attributable to the underestimation of LFR by LIS/OTD within these two regions. Reportedly, the LIS/OTD lightning detection efficiency is highly sensitive to the characteristics of convective clouds (cloud albedo, cloud optical thickness, etc.) (Boccippio et al., 2002; Cecil et al., 2014; email contact with Dr. Daniel J. Cecil (NASA)). High cloud albedo and cloud optical thickness might engender the underestimation of LFR by LIS/OTD. It is also noteworthy that the seasonal variation and long-term trend of global lightning are determined by distinct factors. The seasonal variation of global lightning activity is controlled by the 23° obliquity of Earth's orbit and the asymmetric distribution of the continent between the Northern and Southern hemispheres. However, the long-term global lightning trend we investigated for this study is determined mainly by climate forcers (such as aerosols and GHGs). To minimize the effects of LFR seasonal variation on our study results, we have deseasonalized the results presented in all figures and tables in our revised manuscript by calculating their anomaly based on raw data. The above validation and the deseasonalization of our study results justified that the LFR seasonal variation (and the uncertainties in the simulation of LFR seasonal variation) in our study has a limited effect on the results of this study.

**Referee #2 comment:**

The Pinatubo experiment is the most interesting part of this paper, and I would focus ONLY on this in the revised paper. While this effect is clearly seen in the CMIP6 simulations of lightning, I do not think there has been a specific paper on this topic. Hence, I would encourage the authors to publish a paper only on this, and remove the sections on temperature and aerosols. However, here too, if the model cannot duplicate the climatology of lightning and the annual cycle, then how can we believe the results related to the Pinatubo eruption.

**Author comment:** I appreciate your interest in the findings related to the Pinatubo eruption. Publishing a paper based only on this part (Pinatubo eruption) is a good idea. Nevertheless, for this paper, we still want to show (and it is still valuable to show) our results related to temperature and aerosols. We expect to publish another paper dedicated entirely to the Pinatubo eruption in the future.

Before I explain the issue related to the LFR seasonal variation, I want to mention that **we have optimized the methodology to remove the Pinatubo perturbation** in Stratospheric Aerosol Climatology (SAC). The new methodology can remove the Pinatubo perturbation in SAC more sufficiently and can also maintain the data consistency (please see below, the period 2001–2010 is far from the time of Pinatubo eruption, but 1986-06~1991-05 and 1996-06~2001-05 are closer to the time of Pinatubo eruption). We use the following equations to process SAC **from June 1991 through May 1996**.

$$SAC_{no\_pinatubo} = \begin{cases} SAC_{background}, & |SAC_{raw} - SAC_{background}| > 1.96\sigma, \\ SAC_{raw}, & |SAC_{raw} - SAC_{background}| \leq 1.96\sigma \end{cases}$$

In that equation, $SAC_{no\_pinatubo}$ denotes the stratospheric aerosol climatological data as input data for Volca-off-F1/F2 experiments, $SAC_{background}$ represents the stratospheric background aerosol climatological data (For this study,

$SAC_{background}$ is the corresponding temporal averaged values of the NASA GISS and CCMI stratospheric aerosol dataset **during 1986-06 – 1991-05 and 1996-06 – 2001-05**, when the time is close to the eruption and the stratosphere was less affected by volcanic eruptions). Also, $SAC_{raw}$ stands for the original values of NASA GISS and CCMI stratospheric aerosol dataset during June 1991 – May 1996. Moreover, $\sigma$ symbolizes the standard deviations of stratospheric background aerosol climate data (For this study, $\sigma$ are the corresponding standard deviations of NASA GISS and CCMI stratospheric aerosol dataset during **1986-06 – 1991-05 and 1996-06 – 2001-05**). Instead of using the three-sigma rule, we found that using the 95% confidence interval (1.96$\sigma$) is more appropriate when $SAC_{background}$ and $\sigma$ are calculated based on the data of 1986-06 – 1991-05 and 1996-06 – 2001-05.

Related to the issue of the reproducibility of LFR seasonal variation, as we have described above, the following viewpoints can justify the reliability of our results.

① Both lightning schemes in CHASER can well capture the spatial distribution of LFR in all four seasons when compared to LIS/OTD observations (please see Fig. S1). Both lightning schemes can capture the shifting of lightning distributions clearly along the latitude band in different seasons. Overestimation of LFR by F1/F2 during DJF is primarily attributable to the overestimation of LFR within the Maritime Continent and South America, but this might also be attributable to underestimation of LFR by LIS/OTD within these two regions.

② We have diminished the effects of LFR seasonal variation on our study results. This diminishment has been achieved through the use of deseasonalized data in all figures and tables presented in our revised manuscript (the deseasonalization is conducted by calculating anomalies based on raw data). We have avoided using Figs. 9e, f, which might include the effects of LFR seasonal variation.

**Referee #2 comment:**

Minor comments:

Line 7: "that result in human....."

**Author comment:** We have revised this in the revised manuscript (L7).

**Referee #2 comment:**

Line 18: "Results of sensitivity...."

**Author comment:** We have changed this part of the text in the revised manuscript (L18).

**Referee #2 comment:**

Line 36:   Lightning is not a disaster, it is a hazard

**Author comment:** We have revised this point in the text (L7, L37).

**Referee #2 comment:**

Line 193: I think Pickering et al., 1998 was the first to publish this

Pickering, K.E., Y. Wang, W.K. Tao and C. Price, 1998: Vertical distribution of lightning $NO_x$ for use in regional and global chemical transport models, J. Geophys. Res., 103, 31203-31216

**Author comment:** The paper you have described proposed a "C-shaped" $LNO_x$ vertical profile, but we adopted a new "backward C-shaped" profile, which differs from the "C-shaped" profile.

**Referee #2 comment:**

Line 203: Does the satellite data provide daily data or monthly data? Since the satellite is in LEO (low Earth orbit) orbit, how does it get global data every day?

**Author comment:** The satellite data represent daily data. Because LIS and OTD respectively orbit the Earth 16 times and 14 times a day, they can scan the Earth nearly globally (although some parts still cannot be scanned) every day (Cecil et al., 2014).

**Referee #2 comment:**

Line 313: Why did you stop at 2014? You now have the LIS-ISS data you could also add to the time series.

**Author comment:** Thank you very much for your information. We are also interested in comparing LFR simulations with LIS-ISS observations. However, the CMIP6 forcing datasets (van Marle et al., 2017; Hoesly et al., 2018) for historical experiments stopped in 2014. Some other kinds of forcing datasets provide data after 2014. However, from the perspective of data continuity, the stop time of our simulation is set as 2014.

**Referee #2 comment:**

Line 319: (NCEI) (Fig. 3c)

**Author comment:** We have changed this point in the revised manuscript (L351).

**Referee #2 comment:**

Line 348: Even if this is statistically significant, the r-value is so low that this implies that the temperature and aerosols explain basically nothing of the monthly variability of the lightning activity in the model.

**Author comment:** We used the Mann–Kendall rank statistic test to ascertain whether the effects of global warming on historical lightning trends are statistically significant or not (significance set as 5%). We more explicitly highlighted that we used the Mann–Kendall rank statistic test to conduct the test of significance (L381).

**Referee #2 comment:**

Figure 3: If the aerosol experiment is only a radiative cooling effect, why do we see the largest increase in lightning here? You claim the radiative cooling of aerosols should stabilize the atmosphere. Please explain.

**Author comment:** The Aero1959 experiments exhibit the largest increase in lightning in Fig. 4 is reasonable. As I have described above, the Aero1959 is the experiment with AeroPEs fixed to 1959 throughout the simulation period, not the case with annually varying AeroPEs. During the simulation period, the global aerosol radiative effects increased in STD experiments but remained nearly constant in Aero1959 experiments. Because we simulated almost flat global lightning trends in STD experiments, we can elucidate the increasing global lightning trends simulated using Aero1959 experiments.

**Referee #2 comment:**

Figure 4: Temporal variability of the aerosol forcing and greenhouse forcing should be different, while the response of the lightning is basically the same!

**Author comment:** From our results, the responses to aerosol forcing and GHGs forcing are divergent. As displayed in Fig. 4, global warming engenders an increase in the global lightning trend towards a positive direction, whereas the increases in AeroPEs exert an opposite effect on the global lightning trend.

**Referee #2 comment:**

Line 368: If the CAPE is larger, and the clouds are deeper, why is the volume of precipitating ice decreasing over time. This

does not make physical sense. If the mixed phase region shifts to higher altitudes why would that impact the volume?

**Author comment:**

I very much appreciate your suggestion. Yes, actually "the shifting of 0 – -25°C region to the higher altitude" cannot always explain the reduced $Q_{Ra}$ adequately. Based on an earlier study (Romps, 2019) (please see Fig. 4 in their study), global warming exerts negligible effects on the vertical profile of total solid (cloud ice, snow, and graupel) mass flux ($\varphi$). In other words, $\varphi$ is a roughly invariant function of isotherm under global warming. This phenomenon is also indicated by CHASER's simulation results of our study. However, global warming can engender the increase of CAPE as well as updraft velocity ($w$) within the charge separation regions. $\varphi$ is calculated as a function of the total solid mass mixing ratio ($q$) multiplied by $w$. If $\varphi$ is almost constant but if $w$ increases, then we can expect a decrease of $q$. This finding can explain why $Q_{Ra}$ decreases under global warming. Because the mechanism of the decrease of $Q_{Ra}$ under global warming is not the target of this study, we prefer to cite the relevant literature to explain it (L402–404).

**Referee #2 comment:**

Figure 6: See comments above. Spatial anomalies similar between parameterizations, and similar between forcing of aerosols and greenhouse warming. Most of the changes are in the polar regions, which globally make up a tiny amount of global lightning. Maybe plots with absolute changes would be more informative. I do not see the hatched lines in the top plots.

**Author comment:**

As described above, the past study indicates that climate models simulate similar spatial patterns of global climate response whether the forcing comes from aerosols or GHGs (Kasoar et al., 2018). Based on earlier studies, we infer that the regionally similar spatial patterns of lightning response found in our study would normally be expected in climate model simulations.

We have shown the trends of LFR anomaly in absolute value (fl. km$^{-2}$ yr$^{-2}$) during 1960–2014 on the two-dimensional map in Fig. S4. Actually, the absolute changes (fl. km$^{-2}$ yr$^{-2}$) in LFR anomaly are significant within tropical regions, as displayed in Figs. S4. The hatched lines are also presented in the top plots.

**Referee #2 comment:**

Figure 8: Why are the trends for LNO$_x$ largest for the aerosols experiment?

**Author comment:** We have explained above why lightning trends are largest for Aero1959 experiments. Because we used the same LNO$_x$ production efficiency values for intra-cloud and cloud-to-ground lightning, LFR is linearly related to the LNO$_x$ emissions in our study. We can expect the largest increasing trends of LNO$_x$ emissions for the Aero1959 experiments.

**Referee #2 comment:**

Line 434: Figs. 9a-f

**Author comment:** As described above, we have avoided using Figs. 9e, f, which might include presentation of the effects of LFR seasonal variation. As a result, we cannot see that "simulated global mean LFRs by STD and Volca-off experiments are the same until April 1991" from Figs. 9a–d (because Figs. 9a–d depict the LFR anomaly). Therefore, we have revised this point accordingly (please see L477).

**Referee #2 comment:**

Figure 9: Why do the simulations not return to agree with each other a few years after the Pinatubo eruption when the

aerosols have been removed from the stratosphere? The two maxima in global lightning in 9e is very worrisome, and a cause to reject the paper.

**Author comment:**

Based on earlier studies, the atmosphere is known to be a chaotic system that, although deterministic, is very sensitive to slight perturbations to initial conditions (Zeng et al., 1993). The considerable perturbations caused by the Pinatubo eruption influenced the stratospheric ozone layer and permanently modified the meteorological fields. Such influences can last many years. The simulated lightning cannot return to mutual agreement many years after the Pinatubo eruption because the simulated meteorological fields differ greatly between STD and Volca-off experiments (the Pinatubo eruption permanently modified the meteorological fields). As a chaotic system, the Pinatubo perturbation can be expected to grow over time, rendering the final state unpredictable beyond some future time. Therefore, the peaks of the red line in Fig. 9e can be expected.

**Referee #2 comment:**

Given the reduction of lightning after the Pinatubo eruption, why not calculate the % change per degree C for this event to show the sensitivity of the model lightning to global cooling from a volcano. This may also be of interest to the reader to compare with global warming studies, ENSO studies, solar constant studies, etc.

**Author comment:** Thank you very much for your suggestion. Actually, we found that the decrease in global mean surface temperature after the eruption is not synchronous with the decrease in global mean LFR. This asynchronicity might be attributable to the complex impacts of the enhanced stratospheric aerosol layer on the tropospheric aerosol and cloud microphysics. The decrease in global mean LFR after the eruption is determined by multiple factors (not only the temperature), which makes us difficult to derive the accurate sensitivity (% change per K) for this event. We prefer to examine this issue in our future research.

**Referee #2 comment:**

Figure 13: Why does the blue curve peak in 1991 if this is for the run of Volcano-off??

**Author comment:**

Thank you very much for your question. We displayed the results obtained from STD and STD-rVolcaoff experiments in Fig. 13. It is noteworthy that the experiment setup for STD-rVolcaoff is fundamentally the same as STD experiments (we used the same SSTs/sea ice data and Stratospheric Aerosol Climatology (SAC) for both sets of experiments). The only difference between STD-rVolcaoff and STD experiments is that instead of calculating $LNO_x$ emission rates dynamically by the $LNO_x$ emission scheme, STD-rVolcaoff read the daily $LNO_x$ emission rates calculated from the Volca-off experiments.

As described above, the SSTs/sea ice fields and SAC used for STD-rVolcaoff experiments also contain the signal of Pinatubo perturbation. After the Pinatubo eruption, the decreased SSTs reduced the water vapor in the troposphere. The enhanced stratospheric aerosol layer also reduced the ultraviolet actinic flux in the troposphere. Both of these factors can engender the reduction of OH radical, thereby increasing the methane lifetime. Therefore, the blue curves in Fig. 13 (Fig. 14) also peak in 1991.

**Referee #2 comment:**

Line 565: Wouldn't a linear regression be better than simply using the first and last data point?

**Author comment:** We have used linear regression; the approximating curves are obtained from linear regressions.

**Referee #2 comment:**

Line 587: You do not model the microphysical effects on lightning, so this is misleading. Your study does not simulate the real world effects of aerosols on lightning.

**Author comment:** As we described above, inclusion of the microphysical effects of aerosols on convective clouds might be necessary for better simulation. However, **large uncertainty remains in the microphysical processes in convective clouds** (Tao et al., 2012; Seinfeld et al., 2016; Heikenfeld et al., 2019)**. Therefore, introducing it does not necessarily improve the global lightning distribution and trend**. In addition, the current model ensembles in the latest IPCC/CMIP6 simulations mostly consider CCN only for large-scale (grid-scale) clouds, especially for chemistry–climate models, just as our present study. Only a few models include some kinds of microphysical processes of aerosols for convective (sub-grid-scale) clouds. As also suggested by Referee #1, we consider it is still important to report our experiment results to show possible changes in lightning and $LNO_x$ using one of the state-of-the-art global chemistry climate models.

**Referee #2 comment:**

Line 605: You claim it plausible to see a suppression of lightning due to aerosols, but your simulations show an increase in lightning!!

**Author comment:** The largest increase in lightning trend in Fig. 4 simulated by Aero1959 experiments is reasonable. As I have described above, the Aero1959 experiments used the 1959 AeroPEs throughout simulation period, which was not the case with annually varying AeroPEs. We used the annually varying AeroPEs data (CMIP6 forcing datasets) in the STD experiments, which means that the AeroPEs increased dramatically in the STD experiments. During the simulation period, the global aerosol radiative effects increased in STD experiments but remained nearly constant in Aero1959 experiments. Because the global lightning trends simulated by STD experiments are almost flat, one can expect increasing global lightning trends simulated by Aero1959 experiments. Compared to the lightning trends simulated using Aero1959 experiments, the lightning trends are suppressed by increases in AeroPEs simulated by STD experiments.

**Referee #2 comment:**

In conclusion, I would normally reject such a paper, but allow the authors to make major revisions addressing all my points before consideration of whether to reject or not.

**Author comment:** From the responses to all your concerns above and corresponding revisions made in our revised manuscript, we hope those efforts have improved the manuscript quality and have made it satisfactory for publication.

**References:**

Cecil, D. J., Buechler, D. E., and Blakeslee, R. J.: Gridded lightning climatology from TRMM-LIS and OTD: Dataset description, Atmospheric Research, 135–136, 404–414, https://doi.org/10.1016/j.atmosres.2012.06.028, 2014.

Clark, S. K., Ward, D. S., and Mahowald, N. M.: Parameterization-based uncertainty in future lightning flash density, Geophysical Research Letters, 44, 2893–2901, https://doi.org/10.1002/2017GL073017, 2017.

Heikenfeld, M., White, B., Labbouz, L., and Stier, P.: Aerosol effects on deep convection: the propagation of aerosol perturbations through convective cloud microphysics, Atmospheric Chemistry and Physics, 19, 2601–2627, https://doi.org/10.5194/acp-19-2601-2019, 2019.

Hoesly, R. M., Smith, S. J., Feng, L., Klimont, Z., Janssens-Maenhout, G., Pitkanen, T., Seibert, J. J., Vu, L., Andres, R. J., Bolt, R. M., Bond, T. C., Dawidowski, L., Kholod, N., Kurokawa, J., Li, M., Liu, L., Lu, Z., Moura, M. C. P., O'Rourke, P.

R., and Zhang, Q.: Historical (1750–2014) anthropogenic emissions of reactive gases and aerosols from the Community Emissions Data System (CEDS), Geosci. Model Dev., 11, 369–408, https://doi.org/10.5194/gmd-11-369-2018, 2018.

Holzworth, R. H., Brundell, J. B., McCarthy, M. P., Jacobson, A. R., Rodger, C. J., and Anderson, T. S.: Lightning in the Arctic, Geophysical Research Letters, 48, e2020GL091366, https://doi.org/10.1029/2020GL091366, 2021.

Kasoar, M., Shawki, D., and Voulgarakis, A.: Similar spatial patterns of global climate response to aerosols from different regions, npj Clim Atmos Sci, 1, 1–8, https://doi.org/10.1038/s41612-018-0022-z, 2018.

van Marle, M. J. E., Kloster, S., Magi, B. I., Marlon, J. R., Daniau, A.-L., Field, R. D., Arneth, A., Forrest, M., Hantson, S., Kehrwald, N. M., Knorr, W., Lasslop, G., Li, F., Mangeon, S., Yue, C., Kaiser, J. W., and van der Werf, G. R.: Historic global biomass burning emissions for CMIP6 (BB4CMIP) based on merging satellite observations with proxies and fire models (1750–2015), Geosci. Model Dev., 10, 3329–3357, https://doi.org/10.5194/gmd-10-3329-2017, 2017.

Price, C. and Rind, D.: A simple lightning parameterization for calculating global lightning distributions, Journal of Geophysical Research, 97, 9919–9933, https://doi.org/10.1029/92JD00719, 1992.

Romps, D. M.: Evaluating the Future of Lightning in Cloud-Resolving Models, Geophysical Research Letters, 46, 14863–14871, https://doi.org/10.1029/2019GL085748, 2019.

Rossby, S. A.: Sferics from lightning within a warm cloud, Journal of Geophysical Research (1896–1977), 71, 3807–3809, https://doi.org/10.1029/JZ071i016p03807, 1966.

Seinfeld, J. H., Bretherton, C., Carslaw, K. S., Coe, H., DeMott, P. J., Dunlea, E. J., Feingold, G., Ghan, S., Guenther, A. B., Kahn, R., Kraucunas, I., Kreidenweis, S. M., Molina, M. J., Nenes, A., Penner, J. E., Prather, K. A., Ramanathan, V., Ramaswamy, V., Rasch, P. J., Ravishankara, A. R., Rosenfeld, D., Stephens, G., and Wood, R.: Improving our fundamental understanding of the role of aerosol–cloud interactions in the climate system, Proceedings of the National Academy of Sciences, 113, 5781–5790, https://doi.org/10.1073/pnas.1514043113, 2016.

Takahashi, T.: Electric Charge Life Cycle in Warm Clouds, Journal of the Atmospheric Sciences, 32, 123–142, https://doi.org/10.1175/1520-0469(1975)032<0123:ECLCIW>2.0.CO;2, 1975.

Tao, W. K., Chen, J. P., Li, Z., Wang, C., and Zhang, C.: Impact of aerosols on convective clouds and precipitation, Reviews of Geophysics, 50, 2001, https://doi.org/10.1029/2011RG000369, 2012.

Vaughan, O. H. and Boeck, W. L.: Space Shuttle Video Images: An Example of Warm Cloud Lightning, 1998.

Zeng, X., Pielke Sr, R., and Eykholt, R.: Chaos Theory and Its Applications to the Atmosphere, Bulletin of The American Meteorological Society – Bull Amer Meteorol Soc, 74, 631–644, https://doi.org/10.1175/1520-0477(1993)0742.0.CO, 1993.

---

## Author Response (AR2)

Dear referee, we sincerely appreciate your efforts to review our manuscript and give constructive comments related to it. We have carefully revised our manuscript based on your comments. Hereinafter, I shall respond to your comments point-by-point and explicitly point out changes with line numbers in the revised manuscript. The line number (or Section or Figure number, etc.) in orange is the line number (or Section or Figure number, etc.) in our revised manuscript. Additionally, we use LFR as an abbreviation of the "lightning flash rate".

**Referee #3 comment:**

This paper studies aerosol radiative impact and global warming impacts on lightning within a model. It looks at the multi-decadal trends, the effect of Pinatubo volcano in following years, impacts of lightning $NO_x$ on methane lifetime, and compares to CMIP6 simulations of lightning. The paper concludes that Pinatubo reduced lightning, and that aerosol radiative effect and global warming roughly cancelled out the effect of each other on lightning over the study time period. As such there was little global trend in lightning.

**Author comment:** Many thanks for your dedicated time to review our manuscript.

**Referee #3 comment:**

The paper tries to cover a lot. Due to the spread in analysis directions, the detail is fairly light but generally still to a worthwhile level. The use of both the common cloud top height scheme and an ice-based scheme is commendable, as parametrisation uncertainty is large. My main issue with the paper is the mechanistic explanation of the aerosol impact of lightning trend. I feel the paper plots are more distracting than elucidating. I comment below on how I think that section should be refocused. I don't consider the paper ready to publish until that section is clearer - at the moment the paper would be better without the mechanistic explanation entirely, but some is required.

**Author comment:** We appreciate your affirmation of our manuscript. We also thank you for pointing out its shortcomings. We acknowledge that we can more clearly elucidate the mechanistic explanation of the aerosol impact of lightning trends. We have adopted your suggestions to clarify this issue (please see our comments below).

**Referee #3 comment:**

The Pinatubo result is interesting. It warrants much more focus, but in my mind it is not essential to take it further here for the paper to be publishable. I hope the authors will consider future work modelling both radiative and microphysical impacts of the volcano. There is also then scope for comparing model CAPE and Q to sonde and reanalysis data from the time period to provide observationally robust conclusions.

**Author comment:** Thank you very much for your good suggestions. We also hope to investigate this topic further by considering entire aerosol processes after the volcanic eruption. We agree with you that if we would like to take this topic further, comparing model CAPE and $Q_{Ra}$ with observation-based data is necessary for providing robust conclusions.

**Referee #3 comment:**

Looking at the previous discussion for this paper, other reviewers have been concerned that the study only focuses on aerosol radiative effects, opposed to microphysical effects. This is certainly a limitation of the study, but the authors have now sufficiently acknowledged this, in my opinion. I also consider it useful to have the results published on this topic even with that caveat. Nevertheless, effects on lightning trends are likely to be very different if microphysics is considered and it should be a priority for anyone wanting to take this work further to include that within their scope.

**Author comment:** We appreciate your understanding that our manuscript has not considered the aerosol microphysical effects. As you mentioned, we agree that it is essential to consider aerosol microphysical effects if anyone wants to take this work

further.

**Referee #3 comment:**

______Major comments______

Sec3.2 - The biggest affect on lightning trends is fixing aerosol not climate (fig5). Yet figures have not been presented which show what model factor causes the larger trend in fixed aerosol experiments. Fig4 and fig6 both show larger deviations from the std expt in the fixed climate experiments, and relatively small changes in aero experiments. The figure that should be the basis of understanding this is fig S5 (also S4). This shows that lightning generally increases over tropical land and the warm pool in response to climate trends. It shows that lightning over tropical land generally decreases in response to aerosol radiative trends. We need to have composite changes in CAPE etc from these regions to know why they have the trends they do, and therefore how they affect the global mean response. The existing map plots focus on % changes instead of this absolute change. And as such, are a distraction from the responses that are driving the trends. I note that in fig12 you show absolute changes to describe Pinatubo effects – this is a better approach than in the preceding section. I think it's essential you rethink the narrative of Sec3.2 (most of the analysis is there, except for a few composites maybe).

**Author comment:** We agree with your comments, and we have revised Figs. 6c–6d to show absolute trends of CAPE and $Q_{Ra}$. As also demonstrated in our study (see Fig. 1), most lightning flashes occur over tropical and subtropical land regions. It is displayed in Figs. 6c–6d that the past increases in AeroPEs mostly suppress the CAPE and $Q_{Ra}$ absolute trends within regions with high lightning densities (tropical and subtropical land regions). We further investigated the trends of ±35° latitude land region mean CAPE and $Q_{Ra}$ anomalies, and the results are portrayed in Figs. 6e–6f. Figs. 6e–6f show that past increases in AeroPEs significantly suppress the $Q_{Ra}$ trend (-0.08 % yr$^{-1}$) and slightly suppress the CAPE trend (-0.03 % yr$^{-1}$) within ±35° latitude land regions. Weaker convection activities (smaller CAPE) and fewer hydrometeors (cloud ice, graupel, snow) in the charge separation regions (0°C – -25°C isotherm) engender less lightning. In the case of the ECMWF-McCAUL scheme, CAPE and $Q_{Ra}$ trends were suppressed within ±35° latitude terrestrial regions. This constitutes the main reason for the suppression of the historical global lightning trends induced by increases in AeroPEs through aerosol radiative effects.

We have added the above discussion to our revised manuscript (L433-L441).

**Referee #3 comment:**

Fig2 – what's the confidence interval on the trend estimates? I think these are essential to understand the results. The comment applies throughout the paper to trend estimates.

**Author comment:** Thank you very much for your good suggestion. We have estimated the confidence interval of all trends in our manuscript utilizing Sen's slope estimator (T. et al., 2002; Hussain and Mahmud, 2019). The results are shown in Table 2, Table 3, Table 4, and Table 6.

**Referee #3 comment:**

______Minor comments______

L157 – This is an unusual implementation of price and rind. Please explain how adj_factor is calculated, and reference any evaluations of this implementation. What are the values of adj_factor used?

**Author comment:**

We used a simplified version of equations (Eq. (1) and Eq. (2)) to show the idea that each model layer's cumulus cloud fractions ($Cu\_CF_i$) are used to weight the calculated lightning densities from that layer in the CTH scheme.

$$F_l = \sum_{i=1}^{n=36} adj\_factor \times Cu\_CF_i \times (H_i - H_{surface})^{4.9} \qquad (1)$$

$$F_o = \sum_{i=1}^{n=36} adj\_factor \times Cu\_CF_i \times (H_i - H_{surface})^{1.73} \qquad (2)$$

**The complete equation used in CHASER (MIROC) to implement the CTH scheme is shown below.**

$$FLSHK(IJ) = \sum_{KT=1}^{n=Nlayer} FL0 \times HTOP(IJ,KT)^{FL1} \times \frac{HAREA(IJ) \times LNFRC(IJ,KT) \times CAFCT}{(\frac{CAMAX \times HTOP(IJ,KT)}{CHMAX})^2} \qquad (3)$$

$FLSHK$: the total lightning flash rate from ground to top level at grid IJ.

$IJ$: grid index    $KT$: vertical layer index    $Nlayer$: the number of total vertical layer (here $Nlayer$=36)

$FL0, FL1$: coefficients ($FL0$=3.44 $\times$ $10^{-5}$ and $FL1$=4.9 for land; $FL0$=6.2 $\times$ $10^{-4}$ and $FL1$=1.73 for ocean)

$HTOP$: cloud top height    $HAREA$: grid horizontal area    $LNFRC$: cumulus cloud fraction    $CAFCT$: constant factor (0.43)

$CAMAX$: maximum area of each cumulus ($2.2 \times 10^4$)    $CHMAX$: maximum height of cumulus cloud (12 km)

$HAREA(IJ) \times LNFRC(IJ,KT)$: actual cumulus area at each vertical layer

$(\frac{CAMAX * HTOP(IJ,KT)}{CHMAX})^2$: maximum possible cumulus area at each vertical layer

Terms except $LNFRC(IJ,KT)$ in $\frac{HAREA(IJ) \times LNFRC(IJ,KT) \times CAFCT}{(\frac{CAMAX \times HTOP(IJ,KT)}{CHMAX})^2}$ constitute $adj\_factor$ in Eq. (1) and Eq. (2).

Equation (3) aims to faithfully reflect the contribution of LFR from cumulus with different heights. This form of implementation (Eq. 3) originated from CHASER (Sudo et al., 2002). We have added a citation to our revised manuscript (L154).

**Referee #3 comment:**

Sec3.1 - Both models are poor over the ocean (for different reasons). This should be acknowledged.

**Author comment:** We have acknowledged this point in our revised manuscript (L292-L293).

**Referee #3 comment:**

Fig2 – As far as I can tell, there's no nudging? In which case it would be helpful to note for fig2 that we would not necessarily expect year-to-year variability to be captured, since the only control in the model is SSTs? If there is nudging, this needs to be described.

**Author comment:** Yes, the meteorological nudging was not applied to our simulations. We have noted this point in our revised manuscript (L319-L321).

**Referee #3 comment:**

L344 – Something like the PDO could also be a factor over that length of time.

**Author comment:** Thank you very much for your informative comment. Indeed, there is evidence showing that PDO can also affect lightning activities (Macias Fauria and Johnson, 2006; Mallick et al., 2022). We have added relevant discussion to our revised manuscript (L357-L358).

**Referee #3 comment:**

Fig11 – this would benefit from overplotting the pdf of a year of months following Pinatubo. The distribution could then be tested to show it is different from the climatology.

**Author comment:**

Thanks very much for your good suggestion. We have plotted the PDF of a year of months following Pinatubo in Fig. 11. We have added the relevant discussion to our revised manuscript (L551-L552).

**Reference:**

Hussain, Md. and Mahmud, I.: pyMannKendall: a python package for non parametric Mann Kendall family of trend tests., J. Open Source Softw., 4, 1556, https://doi.org/10.21105/joss.01556, 2019.

Macias Fauria, M. and Johnson, E. A.: Large-scale climatic patterns control large lightning fire occurrence in Canada and Alaska forest regions, J. Geophys. Res. Biogeosciences, 111, https://doi.org/10.1029/2006JG000181, 2006.

Mallick, C., Hazra, A., Saha, S. K., Chaudhari, H. S., Pokhrel, S., Konwar, M., Dutta, U., Mohan, G. M., and Vani, K. G.: Seasonal Predictability of Lightning Over the Global Hotspot Regions, Geophys. Res. Lett., 49, e2021GL096489, https://doi.org/10.1029/2021GL096489, 2022.

Sudo, K., Takahashi, M., Kurokawa, J. I., and Akimoto, H.: CHASER: A global chemical model of the troposphere 1. Model description, J. Geophys. Res. Atmospheres, 107, ACH 7-1-ACH 7-20, https://doi.org/10.1029/2001JD001113, 2002.

T., S., A., M., Anttila, P., Ruoho-Airola, T., and T, A.: Detecting Trends of Annual Values of Atmospheric Pollutants by the Mann-Kendall Test and Sen's Slope Estimates the Excel Template Application MAKESENS, Publ. Air Qual., 31, 2002.

---

## Author Response (AR3)

Dear Editor, we sincerely appreciate your invaluable support and guidance throughout the publication process of our manuscript. We also thank you for accepting our manuscript for publication on ACP.

**Editor's message:**
**Please note one technical correction that is needed: The T. et al. citation (lines 950-952) needs editing to properly include the last names of the first two authors, rather than their initials.**

Author's response:
We have corrected the citation as you mentioned (L936-L938).